# COMPOSITIONAL TRAINING FOR END-TO-END DEEP AUC MAXIMIZATION

**Zhuoning Yuan**[1], **Zhishuai Guo**[1], **Nitesh V. Chawla**[2], **Tianbao Yang**[1]
[1]Department of Computer Science, University of Iowa
[2]Computer Science & Engineering, University of Notre Dame
{zhuoning-yuan, zhishuai-guo, tianbao-yang}@uiowa.edu, nchawla@nd.edu

## ABSTRACT

Recently, deep AUC maximization (DAM) has achieved great success in different domains (e.g., medical image classification). However, the end-to-end training for deep AUC maximization still remains a challenging problem. Previous studies employ an ad-hoc two-stage approach that first trains the network by optimizing a traditional loss (e.g., cross-entropy loss) and then finetunes the network by optimizing an AUC loss. This is because that training a deep neural network from scratch by maximizing an AUC loss usually does not yield a satisfactory performance. This phenomenon can be attributed to the degraded feature representations learned by maximizing the AUC loss from scratch. To address this issue, we propose a novel compositional training framework for end-to-end DAM, namely **compositional DAM**. The key idea of compositional training is to minimize a compositional objective function, where the outer function corresponds to an AUC loss and the inner function represents a gradient descent step for minimizing a traditional loss, e.g., the cross-entropy (CE) loss. To optimize the non-standard compositional objective, we propose an efficient and provable stochastic optimization algorithm. The proposed algorithm enhances the capabilities of both robust feature learning and robust classifier learning by alternatively taking a gradient descent step for the CE loss and for the AUC loss in a systematic way. We conduct extensive empirical studies on imbalanced benchmark and medical image datasets, which unanimously verify the effectiveness of the proposed method. Our results show that the compositional training approach dramatically improves both the feature representations and the testing AUC score compared with traditional deep learning approaches, and yields better performance than the two-stage approaches for DAM as well. The proposed method is implemented in our open-sourced library LibAUC (www.libauc.org) and code is available at https://github.com/Optimization-AI/LibAUC.

## 1 INTRODUCTION

Deep AUC maximization (DAM) represents a new learning paradigm for deep learning, which maximizes the area under ROC curve (AUC) on a training dataset for learning a deep neural network. It has received increasing attention recently due to the advancement in large-scale non-convex optimization algorithms for AUC maximization (Liu et al., 2019a; Yuan et al., 2021; Guo et al., 2020a;b).

Recently, DAM has been successfully applied to different domains with imbalanced data (Yuan et al., 2020; Wang et al., 2021b). For example, Yuan et al. (2020) has employed DAM for a variety of medical image classification tasks, e.g., classification of X-ray images, skin lesion images, mammograms, and microscopic images, and they observed great improvements with about 1%~5% AUC increase over traditional deep learning approaches by optimizing a standard loss function, e.g., the cross-entropy (CE) loss. These pioneering studies on DAM open a new direction for deep learning in the presence of imbalanced data but also raise many questions yet to be solved. A particular question relates to how the network is trained by DAM. Existing studies employ a two-stage approach for DAM, in which the first stage pretrains the network on the training data by optimizing a traditional loss function (e.g. the CE loss) and the second stage finetunes the network by optimizing an AUC loss. It was conjectured in (Yuan et al., 2020) the feature extraction layers learned by directly optimizing AUC from scratch are not as good as optimizing the standard CE loss, similarly as optimizing a class-weighted loss for deep learning with imbalanced data (Cao et al., 2019).

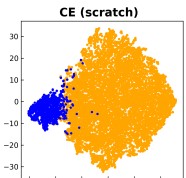 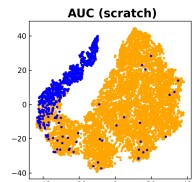 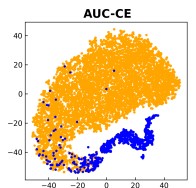 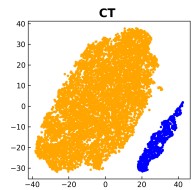

Figure 1: t-SNE visualization of feature representations of an imbalanced training set for the Catvs-Dog visualized by tSNE learned by different methods (from left to right): optimizing CE loss, an AUC loss, a linear combination of CE and AUC loss, and a compositional objective by our method.

Although the ad-hoc two-stage method of DAM has achieved some success, this approach leads to several undesirable consequences increasing the engineering costs in practice: (i) which layers should we finetune in the second stage? Fine-tuning all layers increases the training costs but not necessarily improves the final performance (Jamal et al., 2020; Qi et al., 2020); (ii) when do we stop the training for the first-stage? A long training time for the first-stage increases the overall training costs but not necessarily increases the final prediction performance, while a short training time for the first-stage could harm the prediction performance (Kang et al., 2019). Hence, the literature has suggested different tricks for the two-stage approach, including the decoupling method that simply optimizes the classifier layer in the second-stage (Kang et al., 2019) and deferred re-weighting that only dedicates the iterations with the largest step size to the CE loss (Cao et al., 2019), which could be borrowed to DAM as well. However, an important question remains open regarding DAM:

*How can we conduct end-to-end training for deep AUC maximization?*

To answer this question, we have examined the learned feature representations by optimizing an AUC loss directly from scratch and confirmed the conjecture in (Yuan et al., 2020) that the learned feature representations exhibit no advantage over optimizing the CE loss directly. In Figure 1 we visualize the feature representations on an imbalanced training set for the CatvsDog classification learned by different methods and visualized by t-SNE (van der Maaten & Hinton, 2008). We can see that optimizing an AUC loss from scratch (2nd column) does not yield a cleaner feature representations for the two classes of data than optimizing the CE loss. What makes end-to-end deep learning successful is its superb feature learning capability, i.e., the lower layers capture the low-level features and higher layers capture the high-level features. In terms of feature learning, different examples roughly have equal weights regardless which classes they belong to. From this perspective, we could understand why optimizing AUC loss alone bears worse feature learning capability. The AUC loss assigns different weights to different examples from different classes for more robust classifier learning. In particular, the data in the positive class has a higher weight than data in the negative class. These non-equal weights are important for learning a robust classifier given that feature representations have been learned well but are not readily helpful for learning feature representations in an end-to-end fashion. Can we achieve both effects in a unified and end-to-end learning framework, i.e., optimizing the CE loss with equal weights for robust feature learning and optimizing an AUC loss with uneven weights for robust classifier learning? A naive approach is to simply optimize a linear combination of the CE loss and an AUC loss. However, this approach has a trade-off, meaning that AUC is not necessarily maximized due to the presence of the CE loss in the objective and the learned feature representations could be degraded by the AUC loss (Figure 1, 3rd column).

In this paper, we propose a better and novel end-to-end training method that not only achieves both benefits of minimizing the CE loss for robust feature learning and minimizing an AUC loss for robust classifier learning, but also achieves the effect of "1+1>2", i.e., achieves better performance than the naive linear combination approach. The novel synthesis lies in how we composite the two training steps corresponding to the CE loss and the AUC loss. The central idea is to minimize a two-level compositional objective, where the outer function is an AUC loss, and the inner function is a gradient descent step towards minimizing the CE loss, which represents a quick adaptation to the solution to minimizing the CE loss. We propose a novel efficient stochastic algorithm with provable convergence for minimizing the compositional objective, which performs alternating gradient-based updates that are first based on the gradient of CE loss and then based on the gradient of an AUC loss at the point obtained in the first step. We summarize our contributions below.

• We propose a novel training framework for end-to-end deep AUC maximization, namely compositional DAM. The novel compositional objective enables not only the robust feature learning of

lower layers from minimizing the standard loss function but also the robust learning of a classifier from minimizing an AUC loss.

- Theoretically, we propose an efficient stochastic optimization algorithm for solving compositional DAM, and establish the same convergence rate as standard SGD for optimizing a standard averaged loss. Empirically, we conduct extensive studies on benchmark and medical image datasets, and observe that the proposed method not only improves the baseline methods including the naive linear combination approach but also improves ad-hoc two-stage approaches for DAM. The learned feature representations of compositional DAM (e.g., Figure 1 right) are much better than those learned by minimizing the CE loss or an AUC loss alone and their combination.

## 2 RELATED WORK

**Deep AUC Maximization.** AUC maximization has a history of two decades. Most of the existing studies revolve around the design of efficient optimization algorithms. Earlier papers have focused on full batch methods (Herbrich et al., 1999; Yan et al., 2003; Ferri et al., 2002; Freund et al., 2003; Joachims, 2005; Herschtal & Raskutti, 2004; Rakotomamonjy, 2004; Zhang et al., 2012) and online optimization methods (Zhao et al., 2011; Kar et al., 2014; 2013; Gao et al., 2013). Recently, stochastic optimization for AUC maximization has become the dominating approach (Ying et al., 2016; Liu et al., 2018; Natole et al., 2018; 2019). Ying et al. (2016) propose a milestone work for stochastic optimization of AUC. They consider optimizing the pairwise square loss and propose an equivalent min-max formulation that transforms the original non-decomposable objective into a decomposable one, which enables the design of efficient stochastic methods based on mini-batch of data without explicitly constructing the pairs. The min-max formulation also serves as the basis for recent works on DAM (Liu et al., 2019a; Yuan et al., 2021; Guo et al., 2020a;b). (Liu et al., 2019a) is the first work that explicitly considers DAM and develops the first practical and provable stochastic algorithms for DAM based on the min-max formulation of the pairwise square loss function. However, this work only focuses on optimization and experiments are done on simple benchmark datasets. Later, Yuan et al. (2020) propose a new robust loss in the min-max form for DAM and evaluates the performance of DAM on various medical image classification tasks, which demonstrates great success of DAM. However, none of these works have addressed the problem of end-to-end training for DAM.

**Deep Learning with Imbalanced data.** Deep learning in the presence of imbalanced data has recently attracted tremendous attention (Cui et al., 2019; Johnson & Khoshgoftaar, 2019; Masko & Hensman, 2015; Lee et al., 2016; Khan et al., 2017; Dablain et al., 2021; Ren et al., 2018; Jamal et al., 2020; Qi et al., 2020; Lin et al., 2017; Cao et al., 2019; Kang et al., 2019; Liu et al., 2019b; Zhu & Yang, 2020; Zhou et al., 2020; Xiang et al., 2020; Wang et al., 2021a; 2020; Menon et al., 2021). Among these studies that are closely related to our work include (Lin et al., 2017; Cui et al., 2019; Cao et al., 2019; Qi et al., 2020; Kang et al., 2019; Jamal et al., 2020), which focus on optimizing different objectives from the standard CE loss, including class-weighted loss, focal loss, individually weighted loss functions, etc. Nevertheless, these works are not directly comparable to our method for maximizing AUC.

**Two-stage Approaches.** However, directly optimizing a weighted loss for training a deep neural network from scratch does not work well (Cao et al., 2019; Kang et al., 2019; Jamal et al., 2020; Qi et al., 2020; Yuan et al., 2020). This phenomenon was first observed in (Cao et al., 2019), which is attributed to the degraded feature representations. To tackle this issue, Cao et al. (2019) propose a deferred re-weighting/re-sampling trick. It minimizes the standard average loss for the first stage and switches to minimizing the class-weighted loss or re-sampling method in the second stage. In their paper, the first stage is defined as the training period from the beginning to the iteration that the step size was reduced for the first time in SGD or momentum methods. Kang et al. (2019) investigate a decoupling approach, where the first stage learns the feature representations (i.e., a feature extraction network) by optimizing a standard loss with a large number of iterations, and the second stage learns a robust classifier (i.e. the classifier layer). The authors show that the decoupling approach can achieve better performance than the deferred re-weighting trick in (Cao et al., 2019). However, the decoupling approaches do not necessarily yield the best performance. In particular, some studies have found that fine-tuning some higher layers besides the classifier layer in the second stage is beneficial (Jamal et al., 2020; Qi et al., 2020). Recently, Yang & Xu (2020) propose to use self-supervised learning for learning the feature representations for the first stage and to switch to re-weighting method in the second stage. Different from these studies, our work is to design an elegant end-to-end training framework for deep learning with an AUC loss.

## 3 COMPOSITIONAL TRAINING FOR DEEP AUC MAXIMIZATION

**Notations.** We use $(\mathbf{x}, y)$ to denote an example, where $\mathbf{x} \in \mathbb{R}^{d_0}$ denotes the input and $\mathbf{y} \in \mathcal{Y}$ denotes its corresponding label. Let $\| \cdot \|$ denote the Euclidean norm of a vector, and let $\mathbf{w} \in \mathbb{R}^d$ denote the weight parameters of a deep neural network. A function $F(\mathbf{w})$ is called $L$-smooth if its gradient is $L$-Lipschitz continuous, i.e., $\|\nabla F(\mathbf{w}) - \nabla F(\mathbf{w}')\| \leq L\|\mathbf{w} - \mathbf{w}'\|$. Let $f(\mathbf{w}; \mathbf{x})$ denote the prediction scores of a deep neural network parameterized by $\mathbf{w}$ on an input $\mathbf{x}$, where $f(\mathbf{w}; \mathbf{x}) \in \mathbb{R}$ for binary classification with $\mathcal{Y} = \{1, -1\}$. Denote by $\mathcal{D} = \{(\mathbf{x}_1, y_1), \ldots, (\mathbf{x}_n, y_n)\}$ a set of $n$ training examples. Let $\ell(\mathbf{w}; \mathbf{x}, y)$ denote a loss function on an individual data, e.g., cross-entropy loss. We let $L(\mathbf{w}; \mathcal{S})$ denote an aggregate loss function defined on a set of samples $\mathcal{S} \subseteq \mathcal{D}$. When $\mathcal{S} = \mathcal{D}$, we simply use the notation $L(\mathbf{w}) = L(\mathbf{w}; \mathcal{D})$. Let $\Pi_\Omega[\theta]$ denotes an Euclidean projection on the set $\Omega$. Denote by $n_+(n_-)$ the number of positive (negative) examples.

A standard approach of deep learning is to minimize an averaged loss on training examples, i.e.,

$$\min_{\mathbf{w} \in \mathbb{R}^d} L_{\text{AVG}}(\mathbf{w}) = \frac{1}{n} \sum_{i=1}^{n} \ell(\mathbf{w}; \mathbf{x}_i, y_i). \tag{1}$$

**AUC losses.** AUC (area under the ROC curve) is a commonly used measure for evaluating classifiers for binary classification with imbalanced data. Recently, there emerge voluminous studies on optimizing AUC score for learning a predictive model (e.g., a deep neural network). The idea is to optimize a surrogate loss for the AUC score. A special surrogate loss is the AUC square loss (Gao & Zhou, 2015), which is defined as:

$$\min_{\mathbf{w}} \frac{1}{n_+ n_-} \sum_{y_i=1} \sum_{y_j=-1} (c - (f(\mathbf{w}; \mathbf{x}_i) - f(\mathbf{w}; \mathbf{x}_j)))^2,$$

where $c$ is a margin parameter (e.g., 1). Since directly optimizing the above pairwise loss is computationally expensive, existing works transform the above problem into an equivalent min-max optimization (Liu et al., 2019a), which is decomposable over individual examples:

$$\min_{\mathbf{w}, a, b} \max_{\theta \in \Omega} \Phi(\mathbf{w}, a, b, \theta) := \frac{1}{n} \sum_{i=1}^{n} \phi(\mathbf{w}, a, b, \theta; \mathbf{x}_i, y_i), \quad \text{where} \tag{2}$$

$$\phi(\mathbf{w}, a, b, \theta; \mathbf{x}_i, y_i) = (1 - p)(f(\mathbf{w}; \mathbf{x}_i) - a)^2 \mathbb{I}_{[y_i=1]} + p(f(\mathbf{w}; \mathbf{x}_i) - b)^2 \mathbb{I}_{[y_i=-1]} \tag{3}$$
$$- p(1 - p)\theta^2 + 2\theta \left( p(1 - p)c + pf(\mathbf{w}; \mathbf{x}_i)\mathbb{I}_{[y_i=-1]} - (1 - p)f(\mathbf{w}; \mathbf{x}_i)\mathbb{I}_{[y_i=1]} \right),$$

$\Omega = \mathbb{R}$ and $p = n_+/n$. From the above objective function $\phi$, we can see that each example $\mathbf{x}_i$ also has a class-level weight for their contributed loss, i.e., the data from the positive class is weighted by $1 - p$ and the data from the negative class is weighted by $p$.

It was recently shown that the AUC square loss is sensitive to noisy data and also has adverse effect when trained with easy data. Hence, Yuan et al. (2020) proposed the min-max AUC margin (AUCM) loss, whose optimization problem is (2) with $\Omega = \{0 \leq \theta \leq u\}$ for some $u$. Let us define $L_{\text{AUC}}(\mathbf{w}) = \min_{a,b} \max_{\theta \in \Omega} \Phi(\mathbf{w}, a, b, \theta)$ as the AUC loss function.

### 3.1 COMPOSITIONAL DAM: A COMPOSITIONAL TRAINING METHOD FOR DAM

In this section, we present the proposed compositional training for DAM. Our proposed objective for end-to-end deep learning is given by

$$\min_{\mathbf{w} \in \mathbb{R}^d} L_{\text{AUC}}(\mathbf{w} - \alpha \nabla L_{\text{AVG}}(\mathbf{w})), \tag{4}$$

where $\alpha$ is hyper-parameter. Different from $L_{\text{AUC}}(\mathbf{w})$, the above objective is a compositional function, where the inner component $\mathbf{w} - \alpha \nabla L_{\text{AVG}}(\mathbf{w})$ is another function of $\mathbf{w}$. We refer to the above objective as the compositional objective and a method for minimizing the above compositional objective as compositional training.

To understand the compositional objective (4), we take the second-order Taylor expansion of the compositional objective, which include three terms:

$$L_{\text{AUC}}(\mathbf{w} - \alpha \nabla L_{\text{AVG}}(\mathbf{w})) \approx L_{\text{AUC}}(\mathbf{w}) - \alpha \nabla L_{\text{AUC}}(\mathbf{w})^\top \nabla L_{\text{AVG}}(\mathbf{w}) + C\alpha^2/2\|\nabla L_{\text{AVG}}(\mathbf{w})\|^2, \tag{5}$$

where $C$ represents the Lipchitz continuity constant of $\nabla L_{\text{AUC}}(\cdot)$. In order to understand how the three terms play their roles and evolve in the optimization process by our proposed algorithm presented in next subsection (Algorithm 1), we conduct some empirical studies on several benchmark

Figure 2: Evolution of different terms in (5) computed in the process of our optimization algorithm (Algorithm 1) on the CatvsDog data. $L_{\text{AVG}}$ is the averaged CE loss. Please refer to Appendix A.6 for more details of the calculations.

datasets reported in Appendix A.6. Here, we explain the result of the CatvsDog classification shown in Figure 2. Initially, the first term $L_{\text{AUC}}(\mathbf{w})$ dominates the objective and the algorithm will focus on pushing this term to be smaller (1st column), once it reaches the same level of the third term the algorithm will shift its focus to push $\|\nabla L_{\text{AVG}}(\mathbf{w})\|$ smaller (2nd column) while keeping $\nabla L_{\text{AUC}}(\mathbf{w})^\top \nabla L_{\text{AVG}}(\mathbf{w})$ to be positive (3rd column). This process will continue by alternating between the efforts of pushing $L_{\text{AUC}}(\mathbf{w})$ smaller and of pushing $\|\nabla L_{\text{AVG}}(\mathbf{w})\|^2$ to be smaller while keeping $\nabla L_{\text{AUC}}(\mathbf{w})^\top \nabla L_{\text{AVG}}(\mathbf{w})$ to be non-negative. We also notice that the final AUC loss is close to zero. The same phenomena are also observed on other datasets.

To further understand the compositional training intuitively, let us take a thought experiment by using a gradient descent method to optimize the compositional objective. First, we evaluate the inner function by $\mathbf{u} = \mathbf{w} - \alpha \nabla L_{\text{AVG}}(\mathbf{w})$. We can see that $\mathbf{u}$ is computed by a gradient descent step for minimizing the averaged loss $L_{\text{AVG}}(\mathbf{w})$, which facilitates the learning of lower layers for feature extraction due to equal weights of all examples. Then, we take a gradient descent step to update $\mathbf{w}$ for minimizing the outer function $L_{\text{AUC}}(\cdot)$ by using the gradient $\nabla L_{\text{AUC}}(\mathbf{u})$ instead of $\nabla L_{\text{AUC}}(\mathbf{w})$. Because $\mathbf{u}$ is better than $\mathbf{w}$ in terms of feature extraction layers, taking a gradient descent step using $\nabla L_{\text{AUC}}(\mathbf{u})$ would be better than using $\nabla L_{\text{AUC}}(\mathbf{w})$. In addition, taking a gradient descent step for the outer function $L_{\text{AUC}}(\cdot)$ will make the classifier more robust to the minority class due to the higher weights of examples from the minority class. Overall, we have **two alternating conceptual steps**, i.e., the inner gradient descent step $\mathbf{u} = \mathbf{w} - \alpha \nabla L_{\text{AVG}}(\mathbf{w})$ acts as a **feature purification step**, and the outer gradient descent step $\mathbf{w} - \eta(I - \alpha \nabla^2 L_{\text{AVG}}(\mathbf{w}))\nabla L_{\text{AUC}}(\mathbf{u})$ acts as a **classifier robustification** step, where $\eta$ is a step size.

**VS. linear combination approach.** It is notable that minimizing the compositional objective $L_{\text{AUC}}(\mathbf{w} - \alpha \nabla L_{\text{AVG}}(\mathbf{w}))$ is different from minimizing a linear combination of an AUC loss and the averaged loss, i.e., $L_{\text{AUC}}(\mathbf{w}) + c L_{\text{AVG}}(\mathbf{w})$, where $c > 0$ is a combination weight. First, minimizing the latter objective will push $\nabla L_{\text{AUC}}(\mathbf{w}) + c \nabla L_{\text{AVG}}(\mathbf{w}) = 0$. This makes $\nabla L_{\text{AUC}}(\mathbf{w})$ to have an opposite direction from $\nabla L_{\text{AVG}}(\mathbf{w})$ at the optimal solution, which is different from minimizing the compositional objective in light of the three terms in (5). Second, if we take a gradient descent method for minimizing this objective, the update of $\mathbf{w}$ is given by $\mathbf{w} - \eta(\nabla L_{\text{AUC}}(\mathbf{w}) + c \nabla L_{\text{AVG}}(\mathbf{w}))$. This update is fundamentally different from the two alternating steps of compositional training that use gradients of the AUC loss and the average loss at different points. Third, minimizing the linear combination has a trade-off, meaning that the AUC score is not necessarily maximized due to the presence of the CE loss in the objective and the learned feature representations are degraded due to the presence of AUC loss.

**More discussions.** Finally, we note that the inner gradient descent step $\mathbf{w} - \alpha \nabla L_{\text{AVG}}(\mathbf{w})$ is similar to the idea used in model-agnostic meta learning (MAML) (Finn et al., 2017). However, our compositional objective works fundamentally different from that for MAML. In MAML, the outer loss function and the loss function for the inner gradient descent step is the same. In contrast, the two loss functions in our objective are different. But similar to MAML approaches, we can also run multiple gradient descent steps for the inner function, i.e, the inner function $\mathbf{w} - \alpha \nabla L_{\text{AVG}}(\mathbf{w})$ can be replaced by multiple gradient descent steps. In our experiments, we also found this trick to be helpful for improving the performance.

## 3.2 STOCHASTIC OPTIMIZATION ALGORITHMS

In this subsection, we develop efficient stochastic optimization algorithms for optimizing the compositional objective (4) for DAM. First, we argue the necessity for such development. (i) the problem is a min-max form and the objective is a compositional function, which makes computing an unbiased stochastic gradient of the objective $L_{\text{AUC}}(\mathbf{w} - \alpha \nabla L_{\text{AVG}}(\mathbf{w}))$ impossible. Existing algorithms of

---

**Algorithm 1** Primal-Dual Stochastic Compositional Adaptive (PDSCA) method for solving (6)

---

1: Require Parameters: $\beta_0, \beta_1, \alpha, G_0, \eta_1, \eta_2$
2: Initialization: $\bar{\mathbf{w}}_0 = (\mathbf{w}_0; a_0; b_0) \in \mathbb{R}^{d+2}, \theta_0, \mathbf{u}_0 \in \mathbb{R}^{d+2}$
3: **for** $t = 0, 1, ..., T$ **do**
4:     Sample two sets of examples denoted by $\mathcal{S}_1, \mathcal{S}_2$
5:     $\mathbf{u}_{t+1} = (1 - \beta_0)\mathbf{u}_t + \beta_0 h(\bar{\mathbf{w}}_t; \mathcal{S}_1)$
6:     $\mathcal{O}_t = \nabla_{\bar{\mathbf{w}}} h(\bar{\mathbf{w}}_t; \mathcal{S}_1)^\top [\nabla_{\mathbf{u}} g_1(\mathbf{u}_{t+1}; \mathcal{S}_2) + \theta_t \nabla_{\mathbf{u}} g_2(\mathbf{u}_{t+1}; \mathcal{S}_2)]$
7:     $\mathbf{z}_{t+1} = (1 - \beta_1)\mathbf{z}_t + \beta_1 \mathcal{O}_t$
8:     $\mathbf{z}_{2,t+1} = h_t(\{\mathcal{O}_j, j = 0, \ldots, t\})$                  $\diamond h_t$ can be implemented by that in Appendix B
9:     $\bar{\mathbf{w}}_{t+1} = \bar{\mathbf{w}}_t - \eta_1 \frac{\mathbf{z}_{t+1}}{\sqrt{\mathbf{z}_{2,t+1}} + G_0}$                  $\diamond$with the simplest form $h_t = 1$
10:    $\theta_{t+1} = \Pi_\Omega[\theta_t + \eta_2(g_2(\mathbf{u}_{t+1}; \mathcal{S}_1 \cup \mathcal{S}_2) - \nabla g_3(\theta_t))]$
11: **end for**

---

non-convex min-max optimization for DAM that focus on minimizing $L_{\text{AUC}}(\mathbf{w})$ (Liu et al., 2019a; Yuan et al., 2021; Guo et al., 2020a;b) are not applicable due to the presence of inner function $\mathbf{w} - \alpha \nabla L_{\text{AVG}}(\mathbf{w})$. (ii) Our objective is also different from that of MAML due to that $L_{\text{AUC}}(\cdot)$ is a min-max form, which renders existing algorithms for MAML (Finn et al., 2017; Fallah et al., 2020) not applicable. Hence, below we propose an efficient stochastic algorithm for solving the compositional training for DAM whose objective is of the min-max compositional form, and establish its convergence rate similar to that of standard SGD for minimizing the standard averaged loss.

In particular, for the considered AUC loss, the compositional objective becomes:

$$\min_{\mathbf{w},a,b} \max_{\theta \in \Omega} \Phi\left(\mathbf{w} - \alpha \nabla L_{\text{AVG}}(\mathbf{w}), a, b, \theta\right) = \frac{1}{n} \sum_{i=1}^n \phi\left(\mathbf{w} - \alpha \nabla L_{\text{AVG}}(\mathbf{w}), a, b, \theta; \mathbf{x}_i, y_i\right). \quad (6)$$

We denote by a tuple $\bar{\mathbf{w}} = (\mathbf{w}; a; b)$. For simplicity of presentation, we write $\phi(\mathbf{w}, a, b, \theta, \mathbf{x}_i, y_i)$ as
$$\phi(\bar{\mathbf{w}}, \theta; \mathbf{x}_i, y_i) = g_1(\bar{\mathbf{w}}; \mathbf{x}_i, y_i) + \theta g_2(\bar{\mathbf{w}}; \mathbf{x}_i, y_i) - g_3(\theta),$$
where

$$\begin{aligned}
g_1(\bar{\mathbf{w}}; \mathbf{x}_i, y_i) =& (1 - p)\left(f(\mathbf{w}; \mathbf{x}_i) - a\right)^2 \mathbb{I}_{[y_i=1]} + p(f(\mathbf{w}; \mathbf{x}_i) - b)^2 \mathbb{I}_{[y_i=-1]} \\
& + 2pf(\mathbf{w}; \mathbf{x}_i)\mathbb{I}_{[y_i=1]} - 2(1-p)f(\mathbf{w}; \mathbf{x}_i)\mathbb{I}_{[y_i=-1]},
\end{aligned} \quad (7)$$

and $g_2(\bar{\mathbf{w}}; \mathbf{x}_i, y_i) = 2\left(pf(\mathbf{w}; \mathbf{x}_i)\mathbb{I}_{[y_i=-1]} - (1-p)f(\mathbf{w}; \mathbf{x}_i)\mathbb{I}_{[y_i=1]}\right)$ and $g_3(\theta) = p(1-p)\theta^2$.

Denote by $g_1(\bar{\mathbf{w}}; \mathcal{S}) = \frac{1}{|\mathcal{S}|} \sum_{i \in \mathcal{S}} g_1(\bar{\mathbf{w}}; \mathbf{x}_i, y_i)$, $g_2(\bar{\mathbf{w}}; \mathcal{S}) = \frac{1}{|\mathcal{S}|} \sum_{i \in \mathcal{S}} g_2(\bar{\mathbf{w}}; \mathbf{x}_i, y_i)$. Let $h(\bar{\mathbf{w}}) = (\mathbf{w} - \alpha \nabla L_{\text{AVG}}(\mathbf{w}); a; b)$, $\nabla_{\bar{\mathbf{w}}} h(\bar{\mathbf{w}}) = (I - \alpha \nabla_{\mathbf{w}}^2 L_{\text{AVG}}(\mathbf{w}); 1; 1)$, and $h(\bar{\mathbf{w}}; \mathcal{S}) = (\mathbf{w} - \alpha \nabla L_{\text{AVG}}(\mathbf{w}; \mathcal{S}); a; b)$.

We propose a primal-dual stochastic algorithm shown in Algorithm 1, which is referred to as PDSCA. We provide some explanations of our algorithmic design. First, the step 5 of updating $\mathbf{u}_{t+1}$ corresponds to feature purification step. We use a moving average technique to update $\mathbf{u}_{t+1}$ that takes all historical updates into account, which is inspired by existing stochastic algorithms for optimizing compositional functions (Wang et al., 2017). This is important for us to prove the convergence rate of $O(1/\sqrt{T})$ without using a large batch size at each iteration. If we simply using $\mathbf{u}_{t+1} = h(\bar{\mathbf{w}}_t; \mathcal{S}_1)$ (i.e., setting $\beta_0 = 1$) to estimate $h(\bar{\mathbf{w}}_t)$, there will be a large error in estimating the gradient $\nabla_{\mathbf{u}} g_1(\mathbf{u}_{t+1}; \mathcal{S}_2)$ and $\nabla_{\mathbf{u}} g_2(\mathbf{u}_{t+1}; \mathcal{S}_2)$ in step 6. Second, the step 6 is to estimate the gradient of the outer function. We use two independent mini-batches $\mathcal{S}_1, \mathcal{S}_2$ to ensure that $\mathcal{O}_t$ is an unbiased estimator of $\nabla h(\bar{\mathbf{w}}_t)^\top \nabla_{\bar{\mathbf{w}}} \phi(\mathbf{u}_{t+1}, \theta)$. Using two independent mini-batches is also helpful for improving generalization as demonstrated in experiments. Third, the steps 7 - 9 are similar to the momentum and adaptive methods for updating the model parameter. The step 8 is used for computing the adaptive step size $1/\sqrt{\mathbf{z}_{2,t+1}} + \epsilon_0$, which is similar to adaptive methods used for deep learning, such as Adam, AMSGrad, AdaBound (Kingma & Ba, 2015; Reddi et al., 2018; Luo et al., 2019). We use a general function $h_t$ in the algorithm, which can be implemented by different methods corresponding to different adaptive step size choices. We present different $h_t$ in the Appendix B. The simplest one $h_t = 1$ corresponds to that we do not use adaptive step size and only use the momentum update. Fourth, the step 10 is for updating the dual variable $\theta$ using a stochastic gradient ascent method. Finally, we point out that PDSCA is similar to some existing non-convex strongly-concave min-max optimization algorithms (Guo et al., 2021) but with additional care on the inner gradient descent step $\mathbf{w} - \alpha \nabla L_{\text{AVG}}(\mathbf{w})$. We present an informal convergence of PDSCA below.

**Theorem 1.** *(Informal) Under appropriate conditions on the functions $L_{AVG}$, $g_1$, $g_2$ and a boundness condition on $\theta_t, \mathbf{w}_t, \nabla\ell(\mathbf{w}_t; \mathbf{x}, y), \nabla^2\ell(\mathbf{w}_t; \mathbf{x}, y)$, with $\beta_0, \beta_1 = O(1/\sqrt{T}), \eta_1, \eta_2 = O(1/\sqrt{T})$ and a small constant $\tau$, Algorithm 1 ensures that $\mathbb{E}\left[\frac{1}{T+1}\sum_{t=0}^T \|\nabla F(\bar{\mathbf{w}}_t)\|^2\right] \leq O(\frac{1}{\sqrt{T}})$. where $F(\bar{\mathbf{w}}) = \max_{\theta\in\Omega} \Phi\left(\mathbf{w} - \alpha\nabla L_{AVG}(\mathbf{w}), a, b, \theta\right)$.*

**Remark:** We will present the detailed conditions in the supplement when proving the above theorem due to limit of space. The above theorem indicates that we can optimize the compositional objective (6) with the same convergence rate as optimizing the averaged loss (1) for deep learning.

**Practical Implementations.** It is notable that $\nabla h(\bar{\mathbf{w}}, \mathcal{S}_1) = (I - \alpha\nabla^2 L(\mathbf{w}; \mathcal{S}_1); 1, 1)$ (step 6) involves the Hessian matrix $\nabla^2 L(\mathbf{w}; \mathcal{S}_1)$. Indeed, we only need to compute the Hessian vector product involving in step 6. Similar computation occurs in the meta learning algorithms (Finn et al., 2017; Fallah et al., 2020). Inspired by practical implementations of MAML (Finn et al., 2017) that simply ignore the second-order term, we use the same trick in our experiments. An additional useful trick inspired by MAML is that we can take $k \geq 1$ gradient descent steps for the inner function, correspondingly we maintain and update several $\mathbf{u}$ variables similar to step 5, i.e., using $h(\bar{\mathbf{w}}_t, \mathcal{S}_1)$ for updating the first $\mathbf{u}_{t+1}^{(1)}$, and using $h(\mathbf{u}_{t+1}^{(1)}; \mathcal{S}_1)$ for updating second $\mathbf{u}_{t+1}^{(2)}$, and so on so forth. In our experiments, we found that tuning $k \in \{1, 2, 3\}$ is useful.

Finally, it is notable that although we focus on optimizing AUC loss for binary classification in this work, our compositional training method can be also extended to optimize other weighted losses in an end-to-end fashion, and we include some discussion and results in the Appendix D.

# 4 EXPERIMENTS

In this section, we present some experimental results. We choose five baselines: optimizing the AUC loss from scratch (AUC$^{sc}$), optimizing the CE loss (CE), optimizing a linear combination of the AUC loss and the CE loss with a tuned weight (AUC-CE), the two-stage method with deferred re-weighting trick (Cao et al., 2019) (TS-DRW), the two-stage method by decoupling the learning of feature network by minimizing CE loss and the learning of a classifier by minimizing the AUC loss (TS-DEC) (Kang et al., 2019). We denote our method by CT (AUC). For AUC loss, we use AUCM loss with the margin parameter fixed to be 1 (Yuan et al., 2020). We conduct experiments on four benchmark datasetes and four medical image datasets. The statistics of these datasets are included in the Appendix A.1. More training configurations can be found in Appendix A.2.

**Benchmark datasets.** We choose four benchmark image classification datasets, namely CatvsDog, CIFAR10 (C10), CIFAR100 (C100), and STL10 (S10). For AUC maximization, we construct imbalanced binary versions of these datasets by varying the imbalanced ratios (the ratio of positive examples to the total number of training examples) similar to (Yuan et al., 2020). We use ResNet20 as the prediction network. The weight decay is set to 1e-4 for all experiments. For algorithms to maximize AUC, we use a batch size = 128 and train a total of 100 epochs, and we use step size 0.1 and decrease it by 10 times at 50% and 75% of total training time. We tune the beta parameters of our method in a range $[0.1, 0.99]$ with a grid search and find that good values are around 0.9. For linear combination methods, we tune the weight $c$ of two losses in $\{0.25, 0.5, 0.75\}$. We tune the number of inner gradient steps for CT in $k \in \{1, 2, 3\}$ with $\alpha = 0.1$. For all benchmark data, we run three times for different random seeds and compute the mean and standard deviations.

**Medical image datasets.** We also conduct experiments on naturally imbalanced medical datasets. We choose four medical image datasets, namely Melanoma data, CheXpert, DDSM+, and PatchCam data. The Melanoma dataset is from the Kaggle 2020 competition (Rotemberg et al., 2021), which contains 33,126 labeled images in training set, including 584 positive samples and 32,542 negative samples. We manually construct training, validation and testing datasets following 70/10/20 split. For this dataset, we use the images with 256x256 resolution in the experiments. CheXpert is a large-scale chest X-ray dataset (Irvin et al., 2019), which has 224,316 images with 224 x 224 resolution. The dataset contains 5 binary classification tasks corresponding to 5 diseases, i.e., Cardiomegaly (C0), Edema , Consolidation, Atelectasis, Pleural Effusion. We evaluate the performance on the official validation set consisting of 200 patient studies and report the averaged AUC scores of all 5 diseases. The DDSM+ data is a combination of two datasets namely DDSM and CBIS-DDSM (Lee et al., 2017; Bowyer et al., 1996; Heath et al., 1998), which consists of 55,890 mammographic training images (224×224) with an imratio of 13% and 15,364 images for testing with an imratio of 13%. The PatchCamelyon dataset consists of 294,912 color images (96×96) extracted from

Table 1: Testing performance on benchmark datasets and medical datasets. The percentage number is the second row denotes the imbalanced ratio (cf the text).

| Datasets | | For AUC Maximization | | | Datasets | | For AUC Maximization | |
|---|---|---|---|---|---|---|---|---|
| | imratio | 1% | 10% | 30% | | Method | AUC | |
| CATvsDOG | CE | 0.742±0.003 | 0.917±0.006 | 0.957±0.001 | Melanoma | CE | 0.879±0.008 | |
| | AUC$^{sc}$ | 0.753±0.003 | 0.915±0.002 | 0.964±0.003 | | AUC$^{sc}$ | 0.868±0.006 | |
| | AUC-CE | 0.770±0.007 | 0.939±0.004 | 0.974±0.003 | | AUC-CE | 0.880±0.005 | |
| | TS-DRW | 0.750±0.009 | 0.914±0.003 | 0.961±0.001 | | TS-DRW | 0.878±0.007 | |
| | TS-DEC | 0.754±0.010 | 0.918±0.003 | 0.963±0.001 | | TS-DEC | 0.877±0.005 | |
| | **CT (AUC)** | **0.789±0.008** | **0.946±0.002** | **0.977±0.001** | | **CT (AUC)** | **0.900±0.002** | |
| CIFAR10 | CE | 0.689±0.003 | 0.901±0.002 | 0.944±0.001 | CheXpert | CE | 0.892±0.001 | |
| | AUC$^{sc}$ | 0.728±0.002 | 0.905±0.002 | 0.946±0.001 | | AUC$^{sc}$ | 0.899±0.002 | |
| | AUC-CE | 0.735±0.003 | 0.928±0.001 | 0.957±0.001 | | AUC-CE | 0.902±0.002 | |
| | TS-DRW | 0.708±0.002 | 0.896±0.002 | 0.946±0.003 | | TS-DRW | 0.900±0.002 | |
| | TS-DEC | 0.707±0.002 | 0.897±0.002 | 0.944±0.001 | | TS-DEC | 0.897±0.001 | |
| | **CT (AUC)** | **0.739±0.004** | **0.935±0.001** | **0.964±0.001** | | **CT (AUC)** | **0.909±0.003** | |
| STL10 | CE | 0.655±0.005 | 0.819±0.004 | 0.885±0.004 | DDSM+ | CE | 0.949±0.001 | |
| | AUC$^{sc}$ | 0.665±0.005 | 0.805±0.017 | 0.887±0.007 | | AUC$^{sc}$ | 0.929±0.001 | |
| | AUC-CE | 0.668±0.007 | 0.836±0.006 | 0.905±0.001 | | AUC-CE | 0.957±0.001 | |
| | TS-DRW | 0.655±0.004 | 0.803±0.013 | 0.887±0.002 | | TS-DRW | 0.942±0.003 | |
| | TS-DEC | 0.661±0.002 | 0.816±0.007 | 0.882±0.007 | | TS-DEC | 0.941±0.001 | |
| | **CT (AUC)** | **0.673±0.010** | **0.837±0.006** | **0.906±0.001** | | **CT (AUC)** | **0.981±0.001** | |
| CIFAR100 | CE | 0.586±0.001 | 0.691±0.005 | 0.758±0.004 | PatchCam | CE | 0.869±0.007 | |
| | AUC$^{sc}$ | 0.606±0.004 | 0.705±0.003 | 0.779±0.003 | | AUC$^{sc}$ | 0.868±0.006 | |
| | AUC-CE | 0.605±0.004 | 0.716±0.003 | 0.795±0.001 | | AUC-CE | 0.868±0.005 | |
| | TS-DRW | 0.588±0.002 | 0.691±0.004 | 0.762±0.001 | | TS-DRW | 0.867±0.006 | |
| | TS-DEC | 0.587±0.001 | 0.692±0.003 | 0.762±0.002 | | TS-DEC | 0.869±0.009 | |
| | **CT (AUC)** | **0.609±0.002** | **0.725±0.001** | **0.809±0.002** | | **CT (AUC)** | **0.891±0.003** | |

histopathologic scans of lymph node section for training and 32,768 images for testing with balanced class ratio (Veeling et al., 2018; Bejnordi et al., 2017). For PathCamelyon, we manually construct an imbalanced training dataset with an imratio of 1% and keep the testing set balanced. For Melanoma data, we adopt a EfficientNetV2-S (Tan & Le, 2021) as the network structure, and for CheXpert, DDSM+, PatchCam, we use DenseNet121 (Huang et al., 2017). We tune the number of inner gradient steps for CT in $k \in \{1, 2, 3\}$ and also tune $\alpha$ in $\{0.1, 0.05, 0.01\}$ for the inner steps.

**Results.** The testing AUC results are reported in Table 1. We can see that the proposed composition training method outperforms all baselines on all datasets for maximizing AUC. In addition, we have the following observations: (i) optimizing an AUC loss from scratch does not necessarily yield a better performance than minimizing the standard CE loss; (ii) the CT (AUC) method dramatically improves the performance of optimizing the AUC loss from scratch, with about 2%~5% improvement on difficult medical classification tasks; (iii) the CT (AUC) method is generally better than the linear combination approach (AUC-CE), especially on the more difficult medical image datasets. It is notable that the reported results on some medical datasets are not comparable with that in (Yuan et al., 2020) because (i) we report the performance on official CheXpert validation data instead of the official testing data; (ii) we use a smaller resolution on Melanoma data; (ii) we do not tune the margin parameter in the AUCM loss. We also plot the learned feature representations of training data of different datasets visualized by t-SNE in Figure 3. We can see that the proposed CT (AUC) method obtains better feature representations than the baseline approaches of optimizing CE or AUC alone and the naive linear combination approach.

## 4.1 ABLATION STUDY

We conduct some ablation study including (i) the comparison of convergence curves and the running time analysis of different methods; (ii) the verification of our algorithmic design.

**Convergence Curve.** Below, we compare the convergence speed of our CT (AUC) approach with other end-to-end learning baselines, i.e., CE, AUC, AUC-CE, with results on four benchmark datasets plotted in Figure 4. The results indicate that our algorithm enjoys even faster convergence in terms of number of epochs. The convergence curves of testing AUC are included in Appendix A.4.

**Runtime analysis.** We notice our method has larger running time per-iteration than minimizing the CE and the AUC loss alone because of several backpropagations, but with a reward of faster convergence in epochs and better testing performance. For fair comparison, we have run the baselines with the same amount of time as our method and observed they are still worse than our method (cf Appendix A.5). For example, on CIFAR10 (10%), CT (AUC) can achieve an AUC of 0.944 with a running time of 1000s, in contrast, the baselines CE, AUC, AUC-CE, TS-DRW, and TS-DEC use same running time and achieve AUC scores of 0.925, 0.915, 0.943, 0.917, 0.917, respectively.

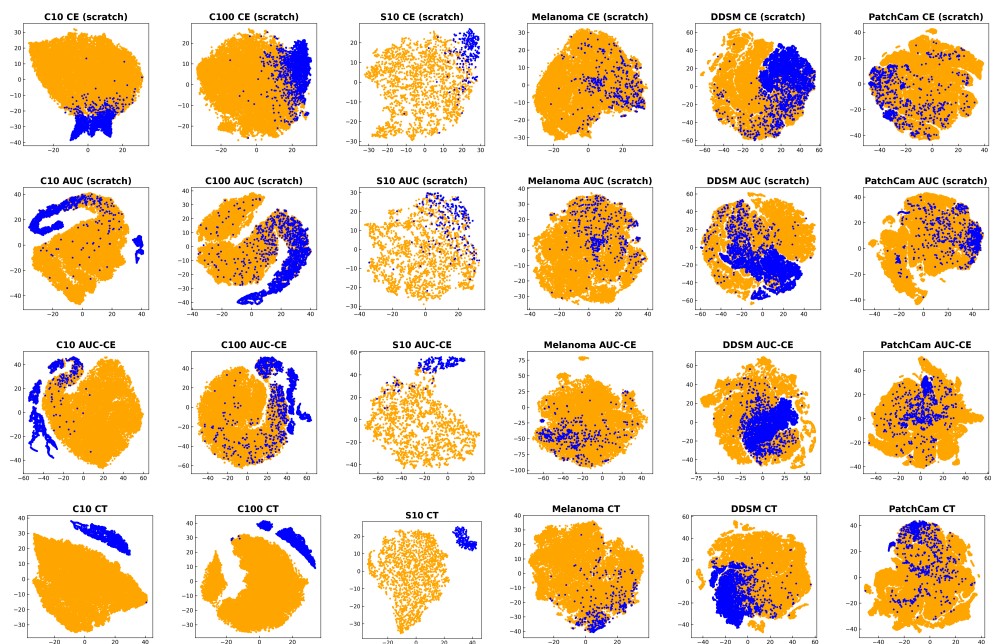

Figure 3: t-SNE visualization of training data (● is positive and ● is negative) by (from top to bottom) optimizing the CE loss, an AUC loss from scratch, a linear combined loss, and our CT method.

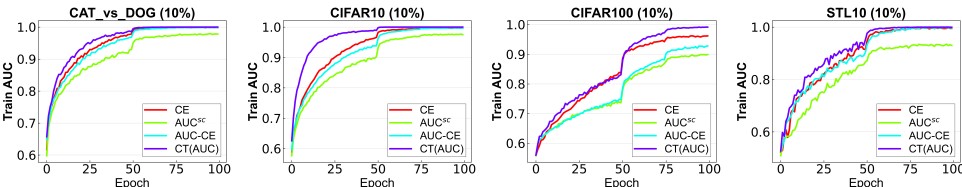

Figure 4: Convergence curves on four benchmark datasets with an imbalance ratio of 10%.

**Verification of Algorithmic Design.** We validate three algorithmic choices. (i) Using two independent mini-batches $\mathcal{S}_1 \neq \mathcal{S}_2$ is generally better than using the same mini-batch. (ii) Using the momentum update for $\mathbf{u}_{t+1}$ (i.e., $\beta_0 < 1$) is better than without using momentum update ($\beta_0 = 1$). (iii) tuning the number of inner gradient steps $k \in \{1, 2, 3\}$ is helpful for improving the performance. The results are demonstrated in Table 2, where all results are averaged over three trials.

Table 2: Left: $\mathcal{S}_1 \neq \mathcal{S}_2$ vs $\mathcal{S}_1 = \mathcal{S}_2$, right: $\beta_0 = 1$ vs $\beta_0 < 1$ in Algorithm 1. Note that we tune $k \in \{1, 2, 3\}$ for for the left table and fix $k = 1$ for the right table. left ($\mathcal{S}_1 \neq \mathcal{S}_2$) vs right ($\beta_0 \leq 1$) verifies that tuning $k$ is helpful.

| Dataset | Method | Imbalance Ratio | | | Method | Imbalance Ratio | | |
|---|---|---|---|---|---|---|---|---|
| | | **1%** | **10%** | **30%** | | **1%** | **10%** | **30%** |
| CATvsDOG | $\mathcal{S}_1 = \mathcal{S}_2$ | 0.784±0.007 | 0.941±0.002 | 0.975±0.001 | $\beta_0 = 1$ | 0.765±0.005 | 0.937±0.004 | 0.971±0.002 |
| | $\mathcal{S}_1 \neq \mathcal{S}_2$ | **0.789±0.008** | **0.946±0.002** | **0.977±0.001** | $\beta_0 < 1$ | **0.769±0.007** | **0.939±0.004** | **0.975±0.006** |
| CIFAR10 | $\mathcal{S}_1 = \mathcal{S}_2$ | 0.738±0.005 | 0.931±0.002 | 0.959±0.000 | $\beta_0 = 1$ | 0.724±0.006 | 0.928±0.004 | 0.957±0.001 |
| | $\mathcal{S}_1 \neq \mathcal{S}_2$ | **0.740±0.004** | **0.935±0.001** | **0.964±0.001** | $\beta_0 < 1$ | **0.725±0.011** | **0.929±0.002** | **0.960±0.001** |
| STL10 | $\mathcal{S}_1 = \mathcal{S}_2$ | 0.671±0.007 | 0.820±0.025 | 0.902±0.003 | $\beta_0 = 1$ | 0.663±0.012 | 0.819±0.018 | 0.895±0.004 |
| | $\mathcal{S}_1 \neq \mathcal{S}_2$ | **0.681±0.005** | **0.839±0.004** | **0.907±0.001** | $\beta_0 < 1$ | **0.666±0.006** | **0.832±0.008** | **0.900±0.002** |
| CIFAR100 | $\mathcal{S}_1 = \mathcal{S}_2$ | 0.608±0.004 | 0.709±0.002 | 0.788±0.005 | $\beta_0 = 1$ | 0.590±0.014 | 0.714±0.004 | 0.791±0.004 |
| | $\mathcal{S}_1 \neq \mathcal{S}_2$ | **0.609±0.002** | **0.725±0.000** | **0.809±0.002** | $\beta_0 < 1$ | **0.598±0.002** | **0.714±0.003** | **0.801±0.001** |

## 5 CONCLUSIONS

In this paper, we have proposed a novel end-to-end compositional training framework for deep AUC maximization by optimizing a compositional objective. We also proposed an efficient stochastic optimization method for compositional training of deep AUC maximization. We demonstrated the effectiveness of compositional training on multiple benchmark datasets and medical datasets for maximizing AUC. In future work, we will investigate compositional training for other imbalanced loss functions more extensively.

ACKNOWLEDGEMENTS

We thank anonymous reviewers for their valuable comments. This work was partially supported by NSF Career Award #1844403, NSF Award #2110545, NSF Award #1933212.

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

## A    Supplement of Experimental Section

### A.1    Dataset Descriptions

The detailed statistics of different datasetse are reported in Table 3. Note that "# of images" refers to the number of samples for the original training set. "LT" denote long-tailed version of the datasets for multi-class tasks. Imbalance ratio (imratio) of XXX (Binary) data means the ratio of number of positive examples to number of all examples. Imbalance ratio (imratio) of XXX (LT) data means ration of the size of smallest class to the size of largest class. The DDSM+ is slightly different from the standard version of CBIS-DDSM or DDSM (Lee et al. (2017); Bowyer et al. (1996); Heath et al. (1998)). We actually use the dataset from (Scuccimarra (2021)) constructed by Eric A. Scuccimarra, which consists of 55k images for training and 15k images for testing. The details about the dataset construction can be found here (Scuccimarra (2021); Zheng et al. (2021); Fulton et al. (2021)). For constructing DDSM+, the positive samples (cancer cases) are from CBIS-DDSM and negative samples (normal cases) are from DDSM. To increase the size of the training data, the author applies offline data augmentation and adds multiple augmented copies to the dataset. In particular, each image (ROI) is randomly cropped three times into 598x598 images, with random flips and rotations, and then the images are resized down to 299x299. For the testing set, the same augmentation is also applied and thus the imbalance ratios remain the same as the training set. The imbalance ratio is about 13% in both training and testing sets. The train/test split in terms of patient ID follows the CBIS-DDSM split, which do not have any overlap. We will add these references in the revision and also correct the citation for CBIS-DDSM. Please also note that this dataset (DDSM+) has also been used for some recent works (Zheng et al. (2021); Fulton et al. (2021)).

Table 3: Description of datasets for classification tasks.

| Dataset | # of images | # of classes | Imbalance Ratio |
|---|---|---|---|
| CATvsDOG (binary) | 20,000 | 2 | 1%, 10%, 30% |
| CIFAR10 (binary) | 50,000 | 2 | 1%, 10%, 30% |
| CIFAR100 (binary) | 50,000 | 2 | 1%, 10%, 30% |
| STL10 (binary) | 5,000 | 2 | 1%, 10%, 30% |
| CheXpert | 223,416 | 2 | 12.2%, 32.2%, 6.8%, 31.2%, 40.3% |
| Melanoma | 33,126 | 2 | 1.76% |
| DDSM+ | 55,000 | 2 | 13% |
| PatchCam | 294,912 | 2 | 1% |
| CIFAR10 (LT) | 50,000 | 10 | 1%, 10% |
| CIFAR100 (LT) | 50,000 | 100 | 1%, 10% |
| STL10 (LT) | 5,000 | 10 | 1%, 10% |
| ImageNet (LT) | 115,800 | 1000 | 0.39% |

### A.2    Training Configurations

All benchmark datasets are experimented by NVIDIA GTX-2080Ti and four medical datasets, i.e., CheXpert, Melanoma, DDSM+ and PatchCam, are experimented by NVIDIA V100. For the datasets of binary classification in Table 1, we use the dataloaders from (Yuan et al., 2020). For the long-tailed datasets of multi-class classification as in Section D, we uses the dataloaders from (Cui et al., 2019). For TS-DRW, we train the models using cross-entropy loss at the first stage and then switch to imbalanced losses at the later stages. For TS-DEC, we first conduct the regular training using cross-entropy loss and same settings and then we discard the trained classifiers and finetune the new classifier for additional 10 epochs using imbalanced loss with the learning rate of 0.01. For the proposed compositional training methods, we consider the non-adaptive version with $h_t(\cdot) = 1$ in all experiments. For all datasets, we use the train/val split to do cross-validation for parameter tuning, except CheXpert as explained below. For the benchmark datasets, we use 19k/1k, 45k/5k, 45k/5k. 4k/1k training/validation split on CatvsDog, CIFAR10, CIFAR100, STL10, respectively. For melanoma dataset, we use 70/10/20 split for train/val/test. For PatchCam, we use their official validation set for tuning parameters, which includes about 37k images with balanced positive and negative samples. For DDSM+, we tune the parameters on 10% data sampled from the training set. For CheXpert, since the official testing set is not released and it will take a long time to evaluate all methods on the official testing data, hence, we evaluate different methods only based on the official validation set with parameters tuned according to this set. To make the experiment on Chexpert con-

sistent with other datasets, we run experiments for all methods on CheXpert by following the same cross-validation procedure, i.e., by sampling 10% training data based on patient ID as the validation set to tune parameters and then we report the average scores of five diseases on the testing set (i.e., the official validation set as a testing set).

## A.3 EVALUATIONS ON MEDICAL DATASET WITH MULTIPLE RUNS

For medical datasets, we run all experiments and report the average performance over three runs. We use batch size of 32 except for PatchCam that is 64, initial learning rate of 0.1 and weight decay of 1e-5. We train Melanoma for 12 epochs, CheXpert for 2 epochs, DDSM+ for 5 epochs and PatchCam for 5 epochs. The learning rate is decayed at 50%, 75% of total training iterations by 10 times. For compositional training, we tune the number of inner gradient steps in $k \in \{1, 2, 3\}$ and also tune $\alpha \in \{0.1, 0.05, 0.01\}$ for the inner steps. The results are summarized in the following table.

Table 4: Testing performance on medical datasets.

| Method | Melanoma | CheXpert | DDSM+ | PatchCam |
|--------|----------|----------|-------|----------|
| | AUC | | | |
| CE | 0.879±0.008 | 0.892±0.001 | 0.949±0.001 | 0.869±0.007 |
| AUC | 0.868±0.006 | 0.899±0.002 | 0.929±0.001 | 0.868±0.006 |
| AUC-CE | 0.880±0.005 | 0.902±0.002 | 0.957±0.001 | 0.868±0.005 |
| TS-DRW | 0.878±0.007 | 0.900±0.002 | 0.942±0.003 | 0.867±0.006 |
| TS-DEC | 0.877±0.005 | 0.897±0.001 | 0.941±0.001 | 0.869±0.009 |
| CT (AUC) | **0.900±0.002** | **0.909±0.003** | **0.981±0.001** | **0.891±0.003** |

## A.4 TESTING CONVERGENCE CURVES.

The convergence curves of testing AUC on the benchmark datasets of different methods are plotted in Figure 5.

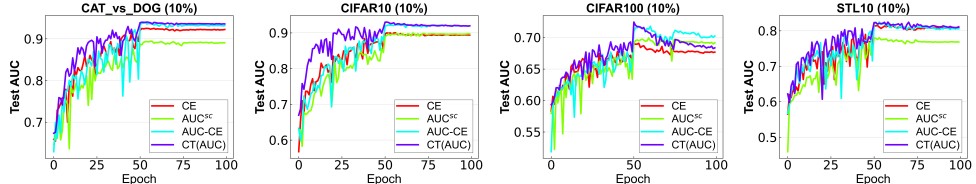

Figure 5: Convergence curves of testing AUC on four benchmark datasets with imbalance ratio of 10%.

## A.5 COMPARISON WITH THE SAME RUNTIME

To compare the performance with the same running time, we train ResNet20 on four benchmark datasets with an imbalance ratio of 10% on a GTX-2080Ti. For each method, we train 1000 seconds and report the best achieved testing AUC. The results are summarized in Table 5.

Table 5: Achieved testing AUC for each method after training ResNet20 for 1000 seconds.

| Dataset | CT (AUC) | CE | AUC | AUC-CE | TS-DRW | TS-DEC |
|---------|----------|-----|-----|--------|--------|--------|
| CATvsDOG (10%) | **0.944** | 0.925 | 0.915 | 0.943 | 0.917 | 0.917 |
| CIFAR10 (10%) | **0.936** | 0.900 | 0.905 | 0.928 | 0.897 | 0.898 |
| STL10 (10%) | **0.837** | 0.815 | 0.783 | 0.829 | 0.816 | 0.815 |
| CIFAR100 (10%) | **0.724** | 0.691 | 0.701 | 0.718 | 0.696 | 0.696 |

## A.6 EVOLUTION OF DIFFERENT TERMS OF THE COMPOSITIONAL OBJECTIVE.

To better understand the proposed compositional objective, we plot the evolution curves of each term in the decomposition equation (5) and compare them with that of naive linear combination approach

(Linear Comb.). We conduct experiments on CATvsDOG, CIFAR10, CIFAR100 and STL10 with imbalance ratio of 10% using ResNet20. We start with initial learning rate of 0.1 and decay it at 50th, 75th epoch by 0.1. In Figure 6, we plot the values of $L_{\text{AUC}}(\mathbf{w})$, $L_{\text{AVG}}(\mathbf{w})$, $\nabla L_{\text{AUC}}(\mathbf{w})^{\top} \nabla L_{\text{AVG}}(\mathbf{w})$, $\|\nabla L_{\text{AVG}}(\mathbf{w})\|^2$ v.s. the number of epochs on training set. For the calculations of $L_{\text{AUC}}(\mathbf{w})$, we compute its values based on the optimal values of $a, b, \alpha$ according to (Yuan et al., 2020) for each epoch. Regarding the calculations of $w$, we compute the mean values of all layers of models. We defer results on other datasets to appendix. From the results, we observe that initially $L_{\text{AUC}}(\mathbf{w})$ dominate the objective and keep decreasing at earlier iterations. When it reaches similar level of the third term $\|\nabla L_{\text{AVG}}(\mathbf{w})\|$, the objective shifts its focus on third term and pushes it to smaller while maintaining second term positive. Eventually, we can see that both AUC and CE losses of CT method reach to a level close to zero. In addition, comparing CT with the linear combination method, it is amazing to see that CT drives both the CE loss and the AUC loss decrease faster and to a smaller level than the linear combination method.

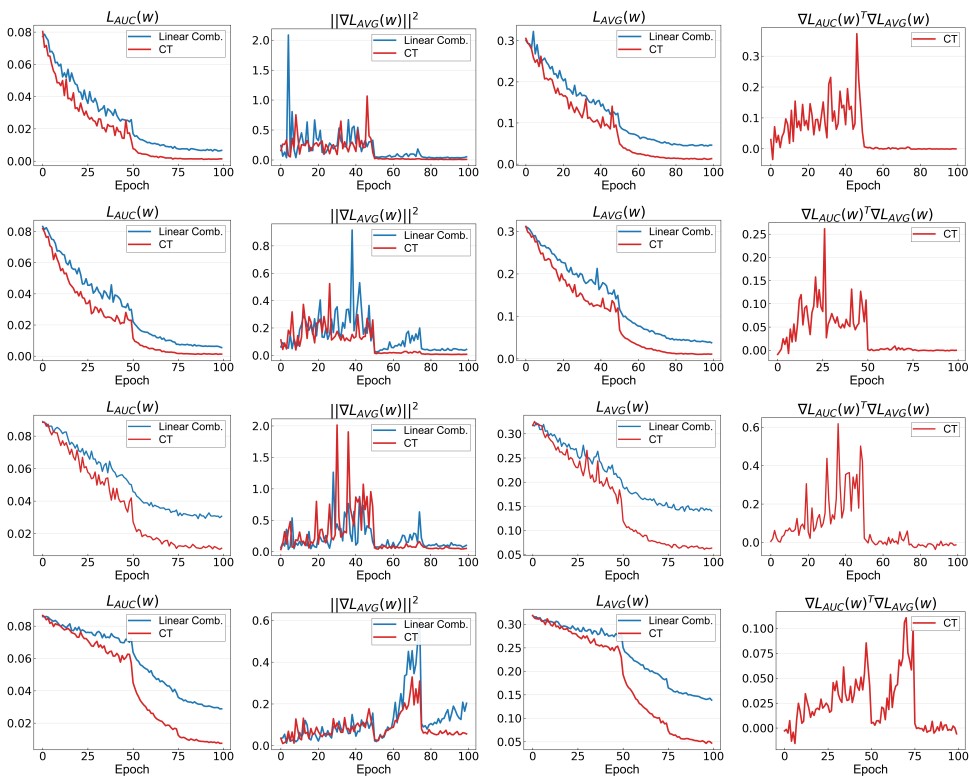

Figure 6: Evolution of each term of compositional objective function on CATvsDog, CIFAR10, STL10 and CIFAR100 datasets (from top to bottom).

# B  IMPLEMENTATION OF $\mathbf{z}_{2,t+1} = h_t(\mathcal{O}_0, \mathcal{O}_1, \ldots, \mathcal{O}_t)$

The $h_t$ is usually implemented by a recursion, where examples are given in Table 6. Similar to (Guo et al., 2021), we make the following assumption for our analysis.

**Assumption 1.** *For the Adam-style algorithms in Table 6, we assume that* $\mathbf{s}_t = 1/(\sqrt{\mathbf{z}_{2,t+1}} + G_0)$ *is upper bounded and lower bounded, i.e., there exists* $0 < c_l < c_u$ *such that* $\forall i, c_l \leq \|\mathbf{s}_{t,i}\| \leq c_u$, *where* $\mathbf{s}_{t,i}$ *denotes the* $i$-*th element of* $\mathbf{s}_t$.

**Remark:** Different implementations of $h_t$ that satisfy this assumption is presented in Table 6.

We need the following lemma to tackle the variance recursion.

**Lemma 1** (Lemma 2, Ghadimi & Wang (2018)). *Consider a moving average sequence* $\mathbf{z}_{t+1} = (1 - \beta)\mathbf{z}_t + \beta_t \mathcal{O}_h(\mathbf{x}_t)$ *for tracking* $h(\mathbf{x}_t)$, *where* $\mathbb{E}[\mathcal{O}_h(\mathbf{x}_t)] = h(\mathbf{x}_t)$ *and* $h$ *is a* $L$-*Lipschitz continuous*

Table 6: Different Adam-style methods and their satisfactions of Assumption 1

| method | update for $h_t$ | Additional assumption | $c_l$ and $c_u$ |
|---|---|---|---|
| SHB | $h_t(\cdot) = 1, G = 0$ | - | $c_l = 1, c_u = 1$ |
| Adam | $\mathbf{z}_{2,t+1} = (1 - \beta'_t)\mathbf{z}_{2,t} + \beta'_t \mathcal{O}_t^2$ | $\|\mathcal{O}_t\|\infty \le G$ | $c_l = \frac{1}{G+G_0}, c_u = \frac{1}{G_0}$ |
| AMSGrad | $\mathbf{z}'_{2,t+1} = (1 - \beta'_t)\mathbf{z}'_{2,t} + \beta'_t \mathcal{O}_t^2$ 
 $\mathbf{z}_{2,t+1} = \max(\mathbf{z}_{2,t}, \mathbf{z}'_{2,t+1})$ | $\|\mathcal{O}_t\|\infty \le G$ | $c_l = \frac{1}{G+G_0}, c_u = \frac{1}{G_0}$ |
| AdaFom (AdaGrad) | $\mathbf{z}_{2,t+1} = \frac{1}{t+1}\sum_{i=0}^{t} \mathcal{O}_t^2$ | $\|\mathcal{O}_t\|\infty \le G$ | $c_l = \frac{1}{G+G_0}, c_u = \frac{1}{G_0}$ |
| Adabound | $\mathbf{z}'_{2,t+1} = (1 - \beta'_t)\mathbf{z}'_{2,t} + \beta'_t \mathcal{O}_t^2$ 
 $\mathbf{z}_{2,t+1} = \Pi_{[1/c_u^2, 1/c_l^2]}[\mathbf{z}'_{2,t+1}], \quad G_0 = 0$ | - | $c_l = c_l, c_u = c_u$ |

*mapping. Then we have*

$$\mathbb{E}_t\|\mathbf{z}_{t+1} - h(\mathbf{x}_t)\|^2 \le (1 - \beta_t)\|\mathbf{z}_t - h(\mathbf{x}_{t-1})\|^2 + 2\beta_t^2 \mathbb{E}_t\|\mathcal{O}_h(\mathbf{x}_t) - h(\mathbf{x}_t)\|^2 + \frac{L^2\|\mathbf{x}_t - \mathbf{x}_{t-1}\|^2}{\beta_t},$$

(8)

*where $\mathbb{E}_t$ denotes the expectation conditioned on all randomness before $\mathcal{O}_h(\mathbf{x}_t)$.*

## C  PROOF OF THEOREM 1

Denote $\eta_t = \eta_1 \mathbf{s}_t$, where $\mathbf{s}_t = 1/(\sqrt{\mathbf{z}_{2,t+1}} + G_0)$. We make the following assumptions regarding problem 6.

**Assumption 2.**

- $\nabla L_{AVG}(\mathbf{w})$ is $C_{L_{AVG}}$-Lipschitz continuous, $g_1(\bar{\mathbf{w}})$ is $C_{g_1}$-Lipschitz continuous and $g_2(\bar{\mathbf{w}})$ is $C_{g_2}$-Lipschitz continuous.

- $\nabla^2 L_{AVG}(\mathbf{w})$ is $L_{L_{AVG}}$-Lipschitz continuous, $\nabla g_1(\bar{\mathbf{w}})$ is $L_{g_1}$-Lipschitz continuous, and $\nabla g_2(\bar{\mathbf{w}})$ is $L_{g_2}$-Lipschitz continuous.

- $\mathbb{E}\|\alpha\nabla L_{AVG}(\mathbf{w}) - \alpha\nabla L_{AVG}(\mathbf{w}; S)\|^2 \le \sigma^2$, $\mathbb{E}\|\alpha\nabla^2 L_{AVG}(\mathbf{w}) - \alpha\nabla^2 L_{AVG}(\mathbf{w}; \mathcal{S})\|^2 \le \sigma^2$, $\mathbb{E}\|\nabla g_1(\mathbf{w}) - \nabla g_1(\mathbf{w}; \mathcal{S})\|^2 \le \sigma^2$, $\mathbb{E}\|g_1(\mathbf{w}) - g_1(\mathbf{w}; \mathcal{S})\|^2 \le \sigma^2$, $\mathbb{E}\|\nabla g_2(\mathbf{w}) - \nabla g_2(\mathbf{w}; \mathcal{S})\|^2 \le \sigma^2$, $\mathbb{E}\|g_2(\mathbf{w}) - g_2(\mathbf{w}; \mathcal{S})\|^2 \le \sigma^2$.

- $\Omega$ is a bounded convex set with radius $D$.

It is notable that the last assumption can be replaced by a condition that the dual variables $\theta_t$ are bounded.

We also know that $g_3(\theta)$ is $\lambda := 2p(1 - p)$-strongly convex and also $L_{g_3} = \lambda$-smooth. Denote $\theta^*(\bar{\mathbf{w}}) = \arg\max_{\theta \in \Omega} \Phi(\bar{\mathbf{w}}, \theta)$. Based on Assumption 2, we have that $h(\bar{\mathbf{w}})$ is $(C_h := 1 + \alpha C_{L_{AVG}})$-Lipschitz continuous, $\nabla h(\bar{\mathbf{w}})$ is $(L_h := 1 + \alpha L_{L_{AVG}})$-Lipschitz continuous, $\mathbb{E}\|\nabla h(\bar{\mathbf{w}}) - \nabla h(\bar{\mathbf{w}}; \mathcal{S})\|^2 \le (1 + D^2)\sigma^2$, and $\mathbb{E}\|h(\bar{\mathbf{w}}) - h(\bar{\mathbf{w}}; \mathcal{S})\|^2 \le (1 + D^2)\sigma^2$. We also know that $F(\bar{\mathbf{w}})$ is $L_F$-smooth, where $L_F := (L_{g_1} + DL_{g_2} + \lambda) + \frac{(L_{g_1} + DL_{g_2} + \lambda)^2}{\lambda}$ (Lemma 4.3 of (Lin et al., 2019)).

We present Theorem 1 formally in the following Theorem.

**Theorem 2.** *Assume $F(\bar{\mathbf{w}}_0) - F_* \le \Delta_F$ where $F_* = \min_{\bar{\mathbf{w}}} F(\bar{\mathbf{w}})$. Suppose Assumptions 1 and 2 hold. With $\beta_0 = O(1/\sqrt{T})$, $\beta_1 = O(1/\sqrt{T})$, $\beta_0 = O(1/\sqrt{T})$, $\beta_1 = O(1/\sqrt{T})$, $\eta_1 \le \min\{\sqrt{\frac{c_l}{c_u^3 C_3}} \frac{\beta_1}{2}, \frac{c_l}{c_u^3(C_1\lambda^2 + 512C_{g_2}^2)} \frac{\beta_0 \lambda}{2C_h}, \frac{c_l}{2c_u^2 L_F}\}$, and $\eta_2 = O(1/\sqrt{T})$ where $C_1$ and $C_3$ are proper constants specified in the proof, Algorithm 1 can ensure that*

$$\mathbb{E}\left[\frac{1}{T+1}\sum_{t=0}^{T} \|\nabla F(\bar{\mathbf{w}}_t)\|^2\right] \le O(\frac{1}{\sqrt{T}}).$$

To prove this theorem, we first need a couple of lemmas.

**Lemma 2.** *Suppose Assumption 1 and Assumption 2 hold. Considering the update in Algorithm 1, we have*

$$F(\bar{\mathbf{w}}_{t+1}) \leq F(\bar{\mathbf{w}}_t) + \frac{\eta_1 c_u}{2}\|[\nabla_h g_1(h(\bar{\mathbf{w}}_t)) + \theta^*(h(\bar{\mathbf{w}}_t))\nabla_h g_2(h(\bar{\mathbf{w}}_t))] - \mathbf{z}_{t+1}\|^2$$
$$- \frac{\eta_1 c_l}{2}\|\nabla F(\bar{\mathbf{w}}_t)\|^2 - \frac{\eta_1 c_l}{4}\|\mathbf{z}_{t+1}\|^2. \tag{9}$$

*Proof of Lemma 2.* Due to the smoothness of $F$, we can prove that under $\eta_1 L_F \leq c_l/(2c_u^2)$,

$$F(\bar{\mathbf{w}}_{t+1}) \leq F(\bar{\mathbf{w}}_t) + \nabla F(\bar{\mathbf{w}}_t)^\top(\bar{\mathbf{w}}_{t+1} - \bar{\mathbf{w}}_t) + \frac{L_F}{2}\|\bar{\mathbf{w}}_{t+1} - \bar{\mathbf{w}}_t\|^2$$

$$= F(\bar{\mathbf{w}}_t) - \nabla F(\bar{\mathbf{w}}_t)^\top(\eta_t \circ \mathbf{z}_{t+1}) + \frac{L_F}{2}\|\eta_t \circ \mathbf{z}_{t+1}\|^2$$

$$= F(\bar{\mathbf{w}}_t) + \frac{1}{2}\|\sqrt{\eta_t} \circ (\nabla F(\bar{\mathbf{w}}_t) - \mathbf{z}_{t+1})\|^2 - \frac{1}{2}\|\sqrt{\eta_t} \circ \nabla F(\bar{\mathbf{w}}_t)\|^2 + (\frac{L_F}{2}\|\eta_t \circ \mathbf{z}_{t+1}\|^2 - \frac{1}{2}\|\sqrt{\eta_t} \circ \mathbf{z}_{t+1}\|^2)$$

$$\leq F(\bar{\mathbf{w}}_t) + \frac{\eta_1 c_u}{2}\|[\nabla_h g_1(h(\bar{\mathbf{w}}_t)) + \theta^*(h(\bar{\mathbf{w}}_t))\nabla_h g_2(h(\bar{\mathbf{w}}_t))] - \mathbf{z}_{t+1}\|^2$$

$$- \frac{\eta_1 c_l}{2}\|\nabla F(\bar{\mathbf{w}}_t)\|^2 + \frac{\eta_1^2 c_u^2 L_F - \eta_1 c_l}{2}\|\mathbf{z}_{t+1}\|^2$$

$$\leq F(\bar{\mathbf{w}}_t) + \frac{\eta_1 c_u}{2}\|[\nabla_h g_1(h(\bar{\mathbf{w}}_t)) + \theta^*(h(\bar{\mathbf{w}}_t))\nabla_h g_2(h(\bar{\mathbf{w}}_t))] - \mathbf{z}_{t+1}\|^2 - \frac{\eta_1 c_l}{2}\|\nabla F(\bar{\mathbf{w}}_t)\|^2 - \frac{\eta_1 c_l}{4}\|\mathbf{z}_{t+1}\|^2.$$
$$\square$$

**Lemma 3.** *Let $\theta_{t+1} = \Pi_\Omega[\theta_t + \eta_2(g_2(\mathbf{u}_{t+1}; \mathcal{S}_1 \cup \mathcal{S}_2) - \nabla g_3(\theta_t))]$, we have*

$$\|\theta_{t+1} - \theta^*(h(\bar{\mathbf{w}}_{t+1}))\|^2 \leq (1 - \frac{\eta_2 \lambda}{2})\mathbb{E}\|\theta_t - \theta^*(h(\bar{\mathbf{w}}_t))\|^2 + 2\eta_2^2\sigma^2$$
$$+ \frac{16\eta_2}{\lambda}C_{g_2}^2\mathbb{E}\|\mathbf{u}_{t+1} - h(\bar{\mathbf{w}}_t)\|^2 + \frac{4L_\theta^2}{\eta_2\lambda}\mathbb{E}\|h(\bar{\mathbf{w}}_t) - h(\bar{\mathbf{w}}_{t+1})\|^2 \tag{10}$$

*where $L_\theta := \frac{C_{g_2}}{\lambda}$ is the Lipschitz continuous constant of $\theta^*(\cdot)$.*

*Proof of Lemma 3.* Since $\theta^*(h(\bar{\mathbf{w}}_t)) = \Pi_\Omega[\theta^*(h(\bar{\mathbf{w}}_t)) + \eta_2(g_2(h(\bar{\mathbf{w}}_t)) - \nabla g_3(\theta^*(h(\bar{\mathbf{w}}_t))))]$, we have

$$\mathbb{E}\|\theta_{t+1} - \theta^*(h(\bar{\mathbf{w}}_t))\|^2$$

$$= \mathbb{E}\|\Pi_\Omega[\theta_t + \eta_2(g_2(\mathbf{u}_{t+1}; \mathcal{S}_1 \cup \mathcal{S}_2) - \nabla g_3(\theta_t))] - \Pi_\Omega[\theta^*(h(\bar{\mathbf{w}}_t)) + \eta_2(g_2(h(\bar{\mathbf{w}}_t)) - \nabla g_3(\theta^*(h(\bar{\mathbf{w}}_t))))]\|^2$$

$$\leq \mathbb{E}\|[\theta_t + \eta_2(g_2(\mathbf{u}_{t+1}; \mathcal{S}_1 \cup \mathcal{S}_2) - \nabla g_3(\theta_t))] - [\theta^*(h(\bar{\mathbf{w}}_t)) + \eta_2(g_2(h(\bar{\mathbf{w}}_t)) - \nabla g_3(\theta^*(h(\bar{\mathbf{w}}_t))))]\|^2$$

$$= \mathbb{E}\|[\theta_t + \eta_2(g_2(\mathbf{u}_{t+1}; \mathcal{S}_1 \cup \mathcal{S}_2) - \nabla g_3(\theta_t)) - \eta_2 g_2(\mathbf{u}_{t+1}) + \eta_2 g_2(\mathbf{u}_{t+1})]$$
$$- [\theta^*(h(\bar{\mathbf{w}}_t)) + \eta_2(g_2(h(\bar{\mathbf{w}}_t)) - \nabla g_3(\theta^*(h(\bar{\mathbf{w}}_t))))]\|$$

$$\leq \mathbb{E}\|[\theta_t + \eta_2(g_2(\mathbf{u}_{t+1}) - \nabla g_3(\theta_t))] - [\theta^*(h(\bar{\mathbf{w}}_t)) + \eta_2(g_2(h(\bar{\mathbf{w}}_t)) - \nabla g_3(\theta^*(h(\bar{\mathbf{w}}_t))))]\|^2$$
$$+ \eta_2^2 \mathbb{E}\|g_2(\mathbf{u}_{t+1}; \mathcal{S}_1 \cup \mathcal{S}_2) - g_2(\mathbf{u}_{t+1})\|^2$$

$$\leq \mathbb{E}\|[\theta_t + \eta_2(g_2(\mathbf{u}_{t+1}) - \nabla g_3(\theta_t))] - [\theta^*(h(\bar{\mathbf{w}}_t)) + \eta_2(g_2(h(\bar{\mathbf{w}}_t)) - \nabla g_3(\theta^*(h(\bar{\mathbf{w}}_t))))]\|^2 + \eta_2^2\sigma^2, \tag{11}$$

where

$$\mathbb{E}\|[\theta_t + \eta_2(g_2(\mathbf{u}_{t+1}) - \nabla g_3(\theta_t))] - [\theta^*(h(\bar{\mathbf{w}}_t)) + \eta_2(g_2(h(\bar{\mathbf{w}}_t)) - \nabla g_3(\theta^*(h(\bar{\mathbf{w}}_t))))]\|^2$$

$$= \mathbb{E}\|\theta_t - \theta^*(h(\bar{\mathbf{w}}_t))\|^2 + \eta_2^2 \mathbb{E}\|[g_2(\mathbf{u}_{t+1}) - \nabla g_3(\theta_t)] - [g_2(h(\bar{\mathbf{w}}_t)) - \nabla g_3(\theta^*(h(\bar{\mathbf{w}}_t)))]\|^2$$
$$+ 2\eta_2\langle\theta_t - \theta^*(h(\bar{\mathbf{w}}_t)), [g_2(\mathbf{u}_{t+1}) - \nabla g_3(\theta_t)] - [g_2(h(\bar{\mathbf{w}}_t)) - \nabla g_3(\theta^*(h(\bar{\mathbf{w}}_t)))]\rangle$$

$$\leq \mathbb{E}\|\theta_t - \theta^*(h(\bar{\mathbf{w}}_t))\|^2 + 2\eta_2^2 C_{g_2}^2 \mathbb{E}\|\mathbf{u}_{t+1} - h(\bar{\mathbf{w}}_t)\|^2 + 2\eta_2^2 L_{g_3}^2 \mathbb{E}\|\theta_t - \theta^*(h(\bar{\mathbf{w}}_t))\|^2 \tag{12}$$

$$+ \frac{\eta_2\lambda}{4}\mathbb{E}\|\theta_t - \theta^*(h(\bar{\mathbf{w}}_t))\|^2 + \frac{4\eta_2}{\lambda}C_{g_2}^2\mathbb{E}\|\mathbf{u}_{t+1} - h(\bar{\mathbf{w}}_t)\|^2 - 2\eta_2\lambda\mathbb{E}\|\theta_t - \theta^*(h(\bar{\mathbf{w}}_t))\|^2$$

$$\leq (1 - \eta_2\lambda)\mathbb{E}\|\theta_t - \theta^*(h(\bar{\mathbf{w}}_t))\|^2 + \frac{8\eta_2}{\lambda}C_{g_2}^2\mathbb{E}\|\mathbf{u}_{t+1} - h(\bar{\mathbf{w}}_t)\|^2,$$

where the first inequality uses strong monotone inequality as $g_3(\cdot)$ is $\lambda$-strongly convex and the second inequality uses $\eta_2 \leq \min\{\frac{\lambda}{8L_{g_3}^2}, \frac{2}{\lambda}\}$. Then,

$$\mathbb{E}\|\theta_{t+1} - \theta^*(h(\bar{\mathbf{w}}_{t+1}))\|^2$$

$$\leq (1 + \frac{\eta_2\lambda}{2})\mathbb{E}\|\theta_{t+1} - \theta^*(h(\bar{\mathbf{w}}_t))\|^2 + (1 + \frac{2}{\eta_2\lambda})\mathbb{E}\|\theta^*(h(\bar{\mathbf{w}}_t)) - \theta^*(h(\bar{\mathbf{w}}_{t+1}))\|^2$$

$$\leq (1 + \frac{\eta_2\lambda}{2})(1 - \eta_2\lambda)\mathbb{E}\|\theta_t - \theta^*(h(\bar{\mathbf{w}}_t))\|^2 + (1 + \frac{\eta_2\lambda}{2})(\eta_2^2\sigma^2 + \frac{8\eta_2}{\lambda}C_{g_2}^2\mathbb{E}\|\mathbf{u}_{t+1} - h(\bar{\mathbf{w}}_t)\|^2)$$

$$+ (1 + \frac{2}{\eta_2\lambda})\mathbb{E}\|\theta^*(h(\bar{\mathbf{w}}_t)) - \theta^*(h(\bar{\mathbf{w}}_{t+1}))\|^2$$

$$\leq (1 + \frac{\eta_2\lambda}{2})(1 - \eta_2\lambda)\mathbb{E}\|\theta_t - \theta^*(h(\bar{\mathbf{w}}_t))\|^2 + 2\eta_2^2\sigma^2 + \frac{16\eta_2}{\lambda}C_{g_2}^2\mathbb{E}\|\mathbf{u}_{t+1} - h(\bar{\mathbf{w}}_t)\|^2$$

$$+ \frac{4L_\theta^2}{\eta_2\lambda}\mathbb{E}\|h(\bar{\mathbf{w}}_t) - h(\bar{\mathbf{w}}_{t+1})\|^2$$

$$\leq (1 - \frac{\eta_2\lambda}{2})\mathbb{E}\|\theta_t - \theta^*(h(\bar{\mathbf{w}}_t))\|^2 + 2\eta_2^2\sigma^2 + \frac{16\eta_2}{\lambda}C_{g_2}^2\mathbb{E}\|\mathbf{u}_{t+1} - h(\bar{\mathbf{w}}_t)\|^2 + \frac{4L_\theta^2}{\eta_2\lambda}\mathbb{E}\|h(\bar{\mathbf{w}}_t) - h(\bar{\mathbf{w}}_{t+1})\|^2,$$

where the third inequality is because that $\theta^*(\cdot)$ is $L_\theta = \frac{C_{g_2}}{\lambda}$-Lipschitz Lin et al. (2019). $\qquad\square$

*Proof of Theorem 2.* Denote by $g_1(h(\bar{\mathbf{w}}_t)) = \mathbb{E}_{\mathbf{x}_i,y_i}[g_1(h(\bar{\mathbf{w}}_t); \mathbf{x}_i, y_i)]$ and $g_2(h(\bar{\mathbf{w}}_t)) = \mathbb{E}_{\mathbf{x}_i,y_i}[g_2(h(\bar{\mathbf{w}}_t); \mathbf{x}_i, y_i)]$. Note that $\nabla_h\Phi(h(\bar{\mathbf{w}}_t), \theta_t) = \nabla_h g_1(h(\bar{\mathbf{w}}_t)) + \theta_t\nabla_h g_2(h(\bar{\mathbf{w}}_t))$. Denote by $\Delta_{u,t} = \|\mathbf{u}_{t+1} - h(\bar{\mathbf{w}}_t)\|^2$, $\Delta_{z,t} = \|\mathbf{z}_{t+1} - \nabla_{\bar{\mathbf{w}}}h(\bar{\mathbf{w}}_t)^\top\nabla_h\Phi(h(\bar{\mathbf{w}}_t), \theta_t)\|^2 = \|\mathbf{z}_{t+1} - \nabla h(\bar{\mathbf{w}}_t)^\top[\nabla_h g_1(h(\bar{\mathbf{w}}_t)) + \theta_t\nabla_h g_2(h(\bar{\mathbf{w}}_t))]\|^2$, and $\delta_t = \|\theta_t - \theta^*(h(\bar{\mathbf{w}}_t))\|^2$.

Applying Lemma 1 to $\mathbf{u}_t$, we have

$$\mathbb{E}[\Delta_{u,t+1}] \leq (1 - \beta_0)\Delta_{u,t} + 2\beta_0^2\sigma^2 + \frac{C_h^2}{\beta_0}\|\bar{\mathbf{w}}_{t+1} - \bar{\mathbf{w}}_t\|^2. \tag{13}$$

Hence we have

$$\mathbb{E}\left[\sum_{t=0}^T \Delta_{u,t}\right] \leq \mathbb{E}\left[\sum_{t=0}^T \frac{\Delta_{u,t} - \Delta_{u,t+1}}{\beta_0} + 2\beta_0\sigma^2(T+1) + \sum_{t=0}^T \frac{C_h^2\eta_1^2 c_u^2\|\mathbf{z}_{t+1}\|^2}{\beta_0^2}\right]. \tag{14}$$

and

$$\mathbb{E}\left[\sum_{t=1}^{T+1} \Delta_{u,t}\right] \leq \mathbb{E}\left[\sum_{t=0}^T \frac{\Delta_{u,t} - \Delta_{u,t+1}}{\beta_0} + 2\beta_0\sigma^2(T+1) + \sum_{t=0}^T \frac{C_h^2\eta_1^2 c_u^2\|\mathbf{z}_{t+1}\|^2}{\beta_0^2}\right]. \tag{15}$$

Define
$$\mathbf{e}_t = (1 - \beta_1)(\nabla_{\bar{\mathbf{w}}}h(\bar{\mathbf{w}}_t)^\top\nabla_h\Phi(h(\bar{\mathbf{w}}_t), \theta^*(h(\bar{\mathbf{w}}_t))) - \nabla_{\bar{\mathbf{w}}}h(\bar{\mathbf{w}}_{t-1})^\top\nabla_h\Phi(h(\bar{\mathbf{w}}_{t-1}), \theta^*(\bar{\mathbf{w}}_{t-1}))).$$

We have
$$\|\mathbf{e}_t\|^2 \leq 2(1-\beta_1)^2[(C_{g_1}^2 + D^2C_{g_2}^2)C_h^2 + C_h^2(L_{g_1}^2 + D^2L_{g_2}^2)](\|\bar{\mathbf{w}}_t - \bar{\mathbf{w}}_{t-1}\|^2 + \|\theta^*(h(\bar{\mathbf{w}}_t)) - \theta^*(h(\bar{\mathbf{w}}_{t-1}))\|^2)$$
and

$$\mathbb{E}\|\mathbf{z}_{t+1} - \nabla_{\bar{\mathbf{w}}}h(\bar{\mathbf{w}}_t)^\top\nabla_h\Phi(h(\bar{\mathbf{w}}_t), \theta^*(h(\bar{\mathbf{w}}_t))) + \mathbf{e}_t\|^2$$

$$\leq \mathbb{E}\|(1 - \beta_1)[\mathbf{z}_t - \nabla_{\bar{\mathbf{w}}}h(\bar{\mathbf{w}}_{t-1})^\top\nabla_h\Phi(h(\bar{\mathbf{w}}_{t-1}), \theta^*(h(\bar{\mathbf{w}}_{t-1})))]$$

$$+ \beta_1[\nabla_{\bar{\mathbf{w}}}h(\bar{\mathbf{w}}_t; \mathcal{S}_1)\nabla_{\mathbf{u}}\Phi(\mathbf{u}_{t+1}, \theta_t; \mathcal{S}_2) - \nabla_{\bar{\mathbf{w}}}h(\bar{\mathbf{w}}_t)^\top\nabla_h\Phi(\mathbf{u}_{t+1}, \theta_t)]$$

$$+ \beta_1[\nabla_{\bar{\mathbf{w}}}h(\bar{\mathbf{w}}_t)^\top\nabla_h\Phi(\mathbf{u}_{t+1}, \theta_t) - \nabla_{\bar{\mathbf{w}}}h(\bar{\mathbf{w}}_t)^\top\nabla_h\Phi(h(\bar{\mathbf{w}}_t), \theta^*(h(\bar{\mathbf{w}}_t)))]\|^2$$

$$\leq \mathbb{E}\|(1 - \beta_1)[\mathbf{z}_t - \nabla_{\bar{\mathbf{w}}}h(\bar{\mathbf{w}}_{t-1})^\top\nabla_h\Phi(h(\bar{\mathbf{w}}_{t-1}), \theta^*(h(\bar{\mathbf{w}}_{t-1})))]$$

$$+ \beta_1[\nabla_{\bar{\mathbf{w}}}h(\bar{\mathbf{w}}_t)^\top\nabla_h\Phi(\mathbf{u}_{t+1}, \theta_t) - \nabla_{\bar{\mathbf{w}}}h(\bar{\mathbf{w}}_t)^\top\nabla_h\Phi(h(\bar{\mathbf{w}}_t), \theta^*(h(\bar{\mathbf{w}}_t)))]\|^2$$

$$+ \beta_1^2\mathbb{E}\|\nabla_{\bar{\mathbf{w}}}h(\bar{\mathbf{w}}_t; \mathcal{S}_1)\nabla_{\mathbf{u}}\Phi(\mathbf{u}_{t+1}, \theta_t; \mathcal{S}_2) - \nabla_{\bar{\mathbf{w}}}h(\bar{\mathbf{w}}_t)^\top\nabla_h\Phi(\mathbf{u}_{t+1}, \theta_t)\|^2$$

$$\leq (1 + \frac{\beta_1}{2})(1 - \beta_1)^2\mathbb{E}\|\mathbf{z}_t - \nabla_{\bar{\mathbf{w}}}h(\bar{\mathbf{w}}_{t-1})^\top\nabla_h\Phi(h(\bar{\mathbf{w}}_{t-1}), \theta^*(h(\bar{\mathbf{w}}_{t-1})))\|^2$$

$$+ (1 + \frac{2}{\beta_1})2\beta_1^2\mathbb{E}\|\nabla_{\bar{\mathbf{w}}}h(\bar{\mathbf{w}}_t)^\top\nabla_h\Phi(\mathbf{u}_{t+1}, \theta_t) - \nabla_{\bar{\mathbf{w}}}h(\bar{\mathbf{w}}_t)^\top\nabla_h\Phi(h(\bar{\mathbf{w}}_t), \theta_t)\|^2$$

$$+ (1 + \frac{2}{\beta_1})2\beta_1^2\mathbb{E}\|\nabla_{\bar{\mathbf{w}}}h(\bar{\mathbf{w}}_t)^\top\nabla_h\Phi(h(\bar{\mathbf{w}}_t), \theta_t) - \nabla_{\bar{\mathbf{w}}}h(\bar{\mathbf{w}}_t)^\top\nabla_h\Phi(h(\bar{\mathbf{w}}_t), \theta^*(h(\bar{\mathbf{w}}_t)))\|^2$$

$$+ \beta_1^2\mathbb{E}\|\nabla_{\bar{\mathbf{w}}}h(\bar{\mathbf{w}}_t; \mathcal{S}_1)^\top\nabla_{\mathbf{u}}\Phi(\mathbf{u}_{t+1}, \theta_t; \mathcal{S}_2) - \nabla_{\bar{\mathbf{w}}}h(\bar{\mathbf{w}}_t)^\top\nabla_h\Phi(\mathbf{u}_{t+1}, \theta_t)\|^2, \tag{16}$$

where the last three terms can be bounded as below. First,

$$\mathbb{E}\|\nabla_{\bar{\mathbf{w}}}h(\bar{\mathbf{w}}_t)^\top \nabla_h\Phi(\mathbf{u}_{t+1},\theta_t) - \nabla_{\bar{\mathbf{w}}}h(\bar{\mathbf{w}}_t)^\top\nabla_h\Phi(h(\bar{\mathbf{w}}_t),\theta_t)\|^2$$
$$\leq 2C_h^2(L_{g_1}^2 + L_{g_2}^2)\mathbb{E}\|\mathbf{u}_{t+1} - h(\bar{\mathbf{w}}_t)\|^2. \tag{17}$$

Second,

$$\mathbb{E}\|\nabla_{\bar{\mathbf{w}}}h(\bar{\mathbf{w}}_t)^\top\nabla_h\Phi(h(\bar{\mathbf{w}}_t),\theta_t) - \nabla_{\bar{\mathbf{w}}}h(\bar{\mathbf{w}}_t)^\top\nabla_h\Phi(h(\bar{\mathbf{w}}_t),\theta^*(h(\bar{\mathbf{w}}_t)))\|^2$$
$$\leq C_h^2 C_{g_2}^2 \mathbb{E}\|\theta_t - \theta^*(h(\bar{\mathbf{w}}_t))\|^2. \tag{18}$$

Third,

$$\mathbb{E}\|\nabla_{\bar{\mathbf{w}}}h(\bar{\mathbf{w}}_t;\mathcal{S}_1)^\top\nabla_{\mathbf{u}}\Phi(\mathbf{u}_{t+1},\theta_t;\mathcal{S}_2) - \nabla_{\bar{\mathbf{w}}}h(\bar{\mathbf{w}}_t)^\top\nabla_h\Phi(\mathbf{u}_{t+1},\theta_t)\|^2$$
$$\leq 8\sigma^2(C_h^2 + D^2C_h^2 + \sigma^2) + 4C_h^2(\sigma^2 + D^2\sigma^2) = 4\sigma^2(3C_h^2 + 3D^2C_h^2 + 2\sigma^2). \tag{19}$$

It follows that

$$\|\mathbf{z}_{t+1} - \nabla_{\bar{\mathbf{w}}}h(\bar{\mathbf{w}}_t)^\top\nabla_h\Phi(h(\bar{\mathbf{w}}_t),\theta^*(h(\bar{\mathbf{w}}_t)))\|^2$$
$$\leq (1+\frac{\beta_1}{2})\mathbb{E}\|\mathbf{z}_{t+1} - \nabla_{\bar{\mathbf{w}}}h(\bar{\mathbf{w}}_t)^\top\nabla_h\Phi(h(\bar{\mathbf{w}}_t),\theta^*(h(\bar{\mathbf{w}}_t))) + \mathbf{e}_t\|^2 + (1+\frac{2}{\beta_1})\|\mathbf{e}_t\|^2$$
$$\leq (1+\frac{\beta_1}{2})^2(1-\beta_1)^2\mathbb{E}\|\mathbf{z}_t - \nabla_{\bar{\mathbf{w}}}h(\bar{\mathbf{w}}_{t-1})\nabla_h\Phi(h(\bar{\mathbf{w}}_{t-1}),\theta_{t-1}))\|^2$$
$$\quad + 32\beta_1 C_h^2(L_{g_1}^2 + L_{g_2}^2)\mathbb{E}\|\mathbf{u}_{t+1} - h(\bar{\mathbf{w}}_t)\|^2 + 16\beta_1\mathbb{E}\|\theta_t - \theta^*(h(\bar{\mathbf{w}}_t))\|^2 \tag{20}$$
$$\quad + 8\beta_1^2\sigma^2(3C_h^2 + 3D^2C_h^2 + 2\sigma^2) + \frac{4}{\beta_1}\|\mathbf{e}_t\|^2$$
$$\leq (1-\beta_1)\mathbb{E}\|\mathbf{z}_t - \nabla_{\bar{\mathbf{w}}}h(\bar{\mathbf{w}}_{t-1})\nabla_h\Phi(h(\bar{\mathbf{w}}_{t-1}),\theta_{t-1}))\|^2 + \beta_1 C_1\mathbb{E}\|\mathbf{u}_{t+1} - h(\bar{\mathbf{w}}_t)\|^2$$
$$\quad + 16\beta_1\mathbb{E}\|\theta_t - \theta^*(h(\bar{\mathbf{w}}_t))\|^2 + \beta_1^2 C_2 + \frac{C_3}{\beta_1}\mathbb{E}\|\bar{\mathbf{w}}_t - \bar{\mathbf{w}}_{t-1}\|^2.$$

where $C_: = 32C_h^2(L_{g_1}^2 + L_{g_2}^2)$, $C_2 := 8\sigma^2(3C_h^2 + 3D^2C_h^2 + 2\sigma^2)$ and $C_3 := 8[(C_{g_1}^2 + D^2C_{g_2}^2)C_h^2 + C_h^2(L_{g_1}^2 + D^2L_{g_2}^2)]$. Thus,

$$\mathbb{E}\left[\sum_{t=0}^T \Delta_{z,t}\right] \leq \mathbb{E}\left[\frac{\Delta_{z,0}}{\beta_1} + C_1\sum_{t=1}^{T+1}\Delta_{u,t} + 16\sum_{t=1}^{T+1}\delta_t + \beta_1 C_2(T+1) + \frac{C_3\eta_1^2 c_u^2}{\beta_1^2}\sum_{t=0}^T\|\mathbf{z}_{t+1}\|^2.\right]$$

Using Lemma 3, we have

$$\sum_{t=0}^T\delta_t \leq \frac{2}{\eta_2\lambda}\mathbb{E}\|\theta_0 - \theta^*(h(\bar{\mathbf{w}}_0))\|^2 + \frac{4\eta_2\sigma^2(T+1)}{\lambda} + \frac{32}{\lambda^2}C_{g_2}^2\sum_{t=0}^T\mathbb{E}[\Delta_{u,t}] + \frac{8L_\theta^2 C_h^2\eta_1^2 c_u^2}{\eta_2^2\lambda^2}\sum_{t=0}^T\mathbb{E}\|\mathbf{z}_{t+1}\|^2,$$

and

$$\sum_{t=1}^{T+1}\delta_t \leq \frac{2}{\eta_2\lambda}\mathbb{E}\|\theta_0 - \theta^*(h(\bar{\mathbf{w}}_0))\|^2 + \frac{4\eta_2\sigma^2(T+1)}{\lambda} + \frac{32}{\lambda^2}C_{g_2}^2\sum_{t=0}^T\mathbb{E}[\Delta_{u,t}] + \frac{8L_\theta^2 C_h^2\eta_1^2 c_u^2}{\eta_2^2\lambda^2}\sum_{t=0}^T\mathbb{E}\|\mathbf{z}_{t+1}\|^2$$

Combining the upper bound of $\sum_t\Delta_{z,t}$, $\sum_t\Delta_{u,t}$, $\sum_t\delta_t$ and Lemma 2, we have

$$\mathbb{E}\left[\sum_{t=0}^T\|\nabla F(\bar{\mathbf{w}}_t)\|^2\right] \leq \frac{2}{\eta_1 c_l}\sum_{t=0}^T\mathbb{E}(F(\bar{\mathbf{w}}_t) - F(\bar{\mathbf{w}}_{t+1})) + \frac{c_u}{c_l}\sum_{t=0}^T\Delta_{z,t} - \frac{1}{2}\sum_{t=0}^T\mathbb{E}\|\mathbf{z}_{t+1}\|^2$$
$$\leq \frac{2\mathbb{E}(F(\bar{\mathbf{w}}_0) - F_*)}{\eta_1 c_l} - \frac{1}{2}\sum_{t=0}^T\mathbb{E}\|\mathbf{z}_{t+1}\|^2 + \frac{c_u}{c_l}\mathbb{E}\left[\frac{\Delta_{z,0}}{\beta_1} + C_1\sum_{t=1}^{T+1}\Delta_{u,t} + 16\sum_{t=1}^{T+1}\delta_t + \beta_1 C_2 + \frac{C_3\eta_1^2 c_u^2}{\beta_1^2}\sum_{t=0}^T\|\mathbf{z}_{t+1}\|^2\right]$$
$$\leq \frac{2\mathbb{E}(F(\bar{\mathbf{w}}_0) - F_*)}{\eta_1 c_l} + \frac{c_u}{c_l}\mathbb{E}\left[\frac{\Delta_{z,0}}{\beta_1} + \frac{32}{\eta_2\lambda}\delta_0 + \beta_1 C_2(T+1) + \frac{64\eta_2\sigma^2(T+1)}{\lambda} + C_1\sum_{t=1}^{T+1}\Delta_{u,t} + \frac{512}{\lambda^2}C_{g_2}^2\sum_{t=0}^T\Delta_{u,t}\right]$$
$$\quad + \sum_{t=0}^T\mathbb{E}\left[\left(\frac{c_u}{c_l}(\frac{C_3\eta_1^2 c_u^2}{\beta_1^2}) - \frac{1}{2}\right)\|\mathbf{z}_{t+1}\|^2\right]$$
$$\leq \frac{2\mathbb{E}(F(\bar{\mathbf{w}}_0) - F_*)}{\eta_1 c_l} + \frac{c_u}{c_l}\mathbb{E}\left[\frac{\Delta_{z,0}}{\beta_1} + \frac{32\delta_0}{\eta_2\lambda} + (C_1 + \frac{512C_{g_2}^2}{\lambda^2})\frac{\Delta_{u,0}}{\beta_0}\right]$$
$$\quad + \frac{c_u(T+1)}{c_l}\mathbb{E}\left[\beta_1 C_2 + \frac{64\eta_2\sigma^2}{\lambda} + 2(C_1 + \frac{512C_{g_2}^2}{\lambda^2})\beta_0\sigma^2\right]$$
$$\quad + \sum_{t=0}^T\mathbb{E}\left[\left(\frac{c_u}{c_l}(\frac{C_3\eta_1^2 c_u^2}{\beta_1^2} + (C_1 + \frac{512C_{g_2}^2}{\lambda^2})\frac{C_h^2\eta_1^2 c_u^2}{\beta_0^2}) - \frac{1}{2}\right)\|\mathbf{z}_{t+1}\|^2\right]$$

Due to the setting

$$\eta_1 \leq \min\{\sqrt{\frac{c_l}{c_u^3 C_3}}\frac{\beta_1}{2}, \frac{c_l}{c_u^3(C_1\lambda^2 + 512C_{g_2}^2)}\frac{\beta_0\lambda}{2C_h}\}, \tag{21}$$

we have

$$\left(\frac{c_u}{c_l}(\frac{C_3\eta_1^2 c_u^2}{\beta_1^2} + (C_1 + \frac{512C_{g_2}^2}{\lambda^2})\frac{C_h^2\eta_1^2 c_u^2}{\beta_0^2}) - \frac{1}{2}\right) \leq 0. \tag{22}$$

Hence, we have

$$\frac{1}{T+1}\mathbb{E}\left[\sum_{t=0}^{T}\|\nabla F(\mathbf{x}_t)\|^2\right] \leq \frac{2\mathbb{E}(F(\bar{\mathbf{w}}_0) - F_*)}{\eta_1 c_l T} + \frac{c_u}{c_l T}\mathbb{E}\left[\frac{\Delta_{z,0}}{\beta_1} + \frac{32\delta_0}{\eta_2\lambda} + (C_1 + \frac{512C_{g_2}^2}{\lambda^2})\frac{\Delta_{u,0}}{\beta_0}\right]$$
$$+ \frac{c_u}{c_l}\mathbb{E}\left[\beta_1 C_2 + \frac{64\eta_2\sigma^2}{\lambda} + 2(C_1 + \frac{512C_{g_2}^2}{\lambda^2})\beta_0\sigma^2\right].$$

With $\beta_0 = O(1/\sqrt{T})$, $\beta_1 = O(1/\sqrt{T})$, and $\eta_2 = O(1/\sqrt{T})$, we have

$$\frac{1}{T+1}\mathbb{E}\left[\sum_{t=0}^{T}\|\nabla F(\mathbf{x}_t)\|^2\right] \leq O(\frac{1}{\sqrt{T}}).$$

$\square$

Table 7: Testing performance on benchmark datasets and ImageNet-LT. The percentage number is the second row denotes the imbalanced ratio (cf the text). All experiments on benchmark datasets are averaged over three runs with different random seeds. The network structure used in all experiments is ResNet32.

| Datasets | For **Accuracy** Maximization | | |
|---|---|---|---|
| | Method | 1% | 10% |
| CIFAR10 (LT) | CE | 0.713±0.001 | 0.876±0.002 |
| | LDAM [Cao et al. (2019)] | 0.744±0.003 | 0.872±0.002 |
| | TS-DRW [Cao et al. (2019)] | 0.780±0.003 | 0.879±0.000 |
| | TS-DEC [Kang et al. (2019)] | 0.758±0.016 | 0.842±0.004 |
| | **CT (CB-LDAM)** | **0.787±0.001** | **0.883±0.001** |
| CIFAR100 (LT) | CE | 0.396±0.002 | 0.572±0.000 |
| | LDAM [Cao et al. (2019)] | 0.407±0.004 | 0.559±0.003 |
| | TS-DRW [Cao et al. (2019)] | 0.427±0.006 | 0.579±0.001 |
| | TS-DEC [Kang et al. (2019)] | 0.403±0.003 | 0.536±0.001 |
| | **CT (CB-LDAM)** | **0.430±0.005** | **0.585±0.002** |
| STL10 (LT) | CE | 0.441±0.017 | 0.639±0.009 |
| | LDAM [Cao et al. (2019)] | 0.440±0.010 | 0.641±0.008 |
| | TS-DRW [Cao et al. (2019)] | 0.458±0.006 | 0.651±0.017 |
| | TS-DEC [Kang et al. (2019)] | 0.457±0.013 | 0.629±0.009 |
| | **CT (CB-LDAM)** | **0.488±0.012** | **0.662±0.005** |
| ImageNet (LT) | CE [Jamal et al. (2020)] | 0.2526 | |
| | CB-CE [Cui et al. (2019)] | 0.2659 | |
| | **CT (CB-CE)** | **0.2661** | |

## D  COMPOSITIONAL TRAINING WITH CLASS WEIGHTED LOSS

In this section, we extend the compositional training method to deep learning with class weighted loss. Let $L_{\text{CW}}(\mathbf{w})$ denote a class weighted loss written as:

$$L_{\text{CW}}(\mathbf{w}) = \frac{1}{n}\sum_{i=1}^{n} p_{y_i}\ell(\mathbf{w}; \mathbf{x}_i, y_i), \tag{23}$$

where $p_{y_i} \in (0, 1)$ denotes a weight assigned to the $i$-th example that depends on the class the data belongs to. There are different methods for determining the class-level weight. A simple method is to set $p_i$ according to the reciprocal of its corresponding class size, i.e., $p_{y_i} = 1/n_{y_i}$. Recently, Cui et al. (2019) proposed an improved variant of class-weighted loss by using an effective number of samples per-class instead of the class size to compute the individual weight, i.e., $p_{y_i} = \frac{1-\gamma}{1-\gamma^{n_{y_i}}}$, where $\gamma \in (0, 1)$ is a hyper-parameter. We refer to the class weighted loss using these individual weights as $L_{\text{CB}}(\mathbf{w}) = 1/n\sum_{i=1}^{n}\frac{1-\gamma}{1-\gamma^{n_{y_i}}}\ell(\mathbf{w}; \mathbf{x}_i, y_i)$.

**With Class-Weighted Losses.** The corresponding compositional objective is a standard two-level compositional function, i.e.,

$$\min_{\mathbf{w} \in \mathbb{R}^d} F(\mathbf{w}) = L_{\text{CW}}(\mathbf{w} - \alpha \nabla L_{\text{AVG}}(\mathbf{w})). \tag{24}$$

Assuming that the stochastic gradient of $L_{\text{CW}}$ can be easily computed, the optimization of the above problem is easier than that for AUC loss.

We present a simplified stochastic adaptive algorithm with an Adam-style adaptive step size in Algorithm 2 referred to as SCA, where $h_t(\mathcal{O}_0, \dots, \mathcal{O}_t)$ denotes an appropriate mapping function, which can be implemented by using different methods, including Adam, AMSGrad, Adabound, etc. Notice that when $h_t(\cdot) = 1$ Algorithm 2 becomes the NASA algorithm (Ghadimi et al., 2020) to stochastic compositional optimization. We present an informal convergence of Algorithm 2 below.

**Theorem 3.** *(Informal) Under appropriate conditions on the loss functions $L_{AVG}$, $L_{CW}$ and $\ell(\mathbf{w}; \mathbf{x}, \mathbf{y})$, with $\beta_0, \beta_1, \eta = O(1/\sqrt{T})$, Algorithm 2 ensures that $\mathbb{E}\left[\frac{1}{T+1}\sum_{t=0}^{T} \|\nabla F(\mathbf{w}_t)\|^2\right] \leq O(\frac{1}{\sqrt{T}})$.*

**Remark:** The appropriate conditions include the Lipschitz continuous conditions on $L_{\text{AVG}}$ and $L_{\text{CW}}$ and bounded variance conditions of $\nabla \ell(\mathbf{w}; \mathbf{x}_i, y_i)$ and $\nabla^2 \ell(\mathbf{w}; \mathbf{x}_i, y_i)$. We will present the detailed conditions below when proving the above theorem. The above theorem indicates that we can optimize the compositional objective (24) with the same convergence rate as optimizing the averaged loss (1) for deep learning.

### D.1 EXPERIMENTS WITH MULTI-CLASS DATASETS

We conduct experiments on three benchmark multi-class image classification datasets, namely CIFAR10, CIFAR100, and STL10. We construct imbalanced versions of these datasets by keep their classes but making the class sizes follow a long-tailed (LT) distribution with two imbalanced ratios (the ratio of the size of minority class to the size of majority class) similar to (Cui et al., 2019). We use ResNet32 as the network stucture. The weight decay is set to 2e-4 for all experiments. For algorithms, we train a total of 200 epochs with a batch size 128 and we use step size 0.1 and decrease it by 10 times at at 80% and 90% of total training time. We tune the beta parameters of our methods in a range $[0.1, 0.99]$ with a grid search and find that good values are around 0.9. For the class-weighted loss, we choose the class-weighted version of the LDAM loss (Cao et al., 2019). For baselines, we compare with optimizing the CE loss (CE), optimizing the LDAM loss (LDAM), the two-stage method with the deferred re-weighting that optimizes the class balanced LDAM loss in the second stage (TS-DRW) (Cao et al., 2019), the two-stage method with the decoupling trick that optimizes the class balanced LDAM loss in the second stage (TS-DEC) (Kang et al., 2019). The results are shown in Table 7. We can see that the proposed CT method achieves the best accuracy on all datasets. In addition, we conduct a large-scale experiment by following Jamal et al. (2020) on ImageNet-LT with ResNet32. We use class-balanced loss (Cui et al., 2019) as outer loss function of our compositional objective. We use an initial learning rate of 0.1 and run a total of 90 epochs decaying learning rate every 35 epochs by a factor of 10. Eventually, we achieve the top1 accuracy of 26.61%, which is better than two baselines, i.e., CE(25.26%) and CBCE(26.59%).

### D.2 ANALYSIS OF THEOREM 3

In the section we analyze Algorithm 2. An algorithm utilizing the adaptive step size and moving average for nonconvex optimization has been studied in (Guo et al., 2021). But here we have to tailor the algorithm and analysis to the considered formulation with composition. Denote $\mathbf{s}_t = 1/(\mathbf{z}_{2,t+1} + G_0)$, $\eta_t = \eta \mathbf{s}_t$. We make the following assumptions regarding the problem 24.

**Assumption 3.**

- $L_{CW}(\mathbf{u})$ is $C_{L_{CW}}$-Lipschitz continuous, $\nabla L_{AVG}(\mathbf{w})$ is $C_{L_{AVG}}$-Lipschitz continuous.

- $\nabla L_{CW}(\mathbf{u})$ is $L_{L_{CW}}$-Lipschitz continuous, $\nabla^2 L_{AVG}(\mathbf{w})$ is $L_{L_{AVG}}$-Lipschitz continuous.

- *The stochastic oracle satisfies* $\mathbb{E}\|\alpha\nabla L_{AVG}(\mathbf{w}) - \alpha\nabla L_{AVG}(\mathbf{w}; S)\|^2 \leq \sigma^2$, $\mathbb{E}\|\alpha\nabla^2 L_{AVG}(\mathbf{w}) - \alpha\nabla^2 L_{AVG}(\mathbf{w}; S)\|^2 \leq \sigma^2$, $\mathbb{E}\|\nabla L_{CW}(\mathbf{u}) - \nabla L_{CW}(\mathbf{u}; S)\|^2 \leq \sigma^2$.

We formally present Theorem 3 as follows

---

**Algorithm 2** Stochastic Compositional Adaptive (SCA) method for solving (24)

1: Require Parameters: $\beta_0, \beta_1, \alpha, G_0, \eta$
2: Initialization: $\mathbf{w}_0 \in \mathbb{R}^d, \mathbf{z}_0, \mathbf{u}_0$
3: **for** $t = 0, 1, ..., T$ **do**
4:     Sample three sets of examples denoted by $\mathcal{S}_1, \mathcal{S}_2$
5:     $\mathbf{u}_{t+1} = (1 - \beta_0)\mathbf{u}_t + \beta_0(\mathbf{w}_t - \alpha\nabla L_{\text{AVG}}(\mathbf{w}_t; \mathcal{S}_1))$
6:     $\mathcal{O}_t = (I - \alpha\nabla^2 L_{\text{AVG}}(\mathbf{w}_t; \mathcal{S}_1))\nabla L_{\text{CW}}(\mathbf{u}_{t+1}; \mathcal{S}_2)$
7:     $\mathbf{z}_{t+1} = (1 - \beta_1)\mathbf{z}_t + \beta_1 \mathcal{O}_t$
8:     $\mathbf{z}_{2,t+1} = h_t(\mathcal{O}_0, \ldots, \mathcal{O}_t)$           $\diamond h_t$ can be implemented by that in Appendix B,
9:     $\mathbf{w}_{t+1} = \mathbf{w}_t - \eta\frac{\mathbf{z}_{t+1}}{\sqrt{\mathbf{z}_{2,t+1}}+G_0}$           $\diamond$with the simplest form $h_t = 1$
10: **end for**

---

**Theorem 4.** *Assume $F(\mathbf{x}_0) - F_* \leq \Delta_F$ where $F_* = \min\limits_{\mathbf{x}} F(\mathbf{x})$. Suppose Assumptions 1 and 3 hold. With $\eta \leq \left\{ \frac{\sqrt{c_l}\beta_1}{4L_F\sqrt{c_u^3}}, \frac{\sqrt{c_l}\beta_0}{4(1+\alpha C_{L_{\text{AVG}}})\sqrt{C_5 c_u^3}}, \frac{c_l}{2c_u^2 L_F} \right\}, \beta_0 = O(\frac{1}{\sqrt{T}}), \beta_1 \leq O(\frac{1}{\sqrt{T}})$, and constants $L_F = 2L_{L_{\text{CW}}}(1 + \alpha C_{L_{\text{AVG}}})^2 + 2C_{L_{\text{CW}}}\alpha L_{L_{\text{AVG}}}, C_5 = (4L_{L_{\text{AVG}}}^2 C_{L_{\text{CW}}}^2 + 2(1 + \alpha L_{L_{\text{AVG}}}^2)L_{L_{\text{CW}}})$, Algorithm 2 can ensure that*

$$\mathbb{E}\left[ \frac{1}{T+1}\sum_{t=0}^{T} \|\nabla F(\mathbf{w}_t)\|^2 \right] \leq O(\frac{1}{\sqrt{T}}).$$

*Proof of Theorem 4.* By Assumption 2, we know that $F(\mathbf{w})$ is smooth with coefficient $L_F := 2L_{L_{\text{CW}}}(1 + \alpha C_{L_{\text{AVG}}})^2 + 2C_{L_{\text{CW}}}\alpha L_{L_{\text{AVG}}}$. We can prove that under $\eta L_F \leq c_l/(2c_u^2)$,

$$F(\mathbf{w}_{t+1}) \leq F(\mathbf{w}_t) + \nabla F(\mathbf{w}_t)^\top(\mathbf{w}_{t+1} - \mathbf{w}_t) + \frac{L_F}{2}\|\mathbf{w}_{t+1} - \mathbf{w}_t\|^2$$

$$= F(\mathbf{w}_t) - \nabla F(\mathbf{w}_t)^\top(\eta_t \circ \mathbf{z}_{t+1}) + \frac{L_F}{2}\|\eta_t \circ \mathbf{z}_{t+1}\|^2$$

$$= F(\mathbf{w}_t) + \frac{1}{2}\|\sqrt{\eta_t} \circ (\nabla F(\mathbf{w}_t) - \mathbf{z}_{t+1})\|^2 - \frac{1}{2}\|\sqrt{\eta_t} \circ \nabla F(\mathbf{w}_t)\|^2 + (\frac{L_F}{2}\|\eta_t \circ \mathbf{z}_{t+1}\|^2 - \frac{1}{2}\|\sqrt{\eta_t} \circ \mathbf{z}_{t+1}\|^2)$$

$$\leq F(\mathbf{w}_t) + \frac{\eta c_u}{2}\|\nabla F(\mathbf{w}_t) - \mathbf{z}_{t+1}\|^2 - \frac{\eta c_l}{2}\|\nabla F(\mathbf{w}_t)\|^2 + \frac{\eta^2 c_u^2 L_F - \eta c_l}{2}\|\mathbf{z}_{t+1}\|^2$$

$$\leq F(\mathbf{w}_t) + \frac{\eta c_u}{2}\|\nabla F(\mathbf{w}_t) - \mathbf{z}_{t+1}\|^2 - \frac{\eta c_l}{2}\|\nabla F(\mathbf{w}_t)\|^2 - \frac{\eta c_l}{4}\|\mathbf{z}_{t+1}\|^2.$$

$$(25)$$

Denote $\Delta_{z,t} = \|\mathbf{z}_{t+1} - \nabla F(\mathbf{w}_t)\| = \|\mathbf{z}_{t+1} - (I - \alpha\nabla^2 L_{\text{AVG}}(\mathbf{w}_t))\nabla L_{\text{CW}}(\mathbf{w}_t - \alpha\nabla L_{\text{AVG}}(\mathbf{w}_t))\|^2$ and $\Delta_{u,t} = \|\mathbf{u}_{t+1} - (\mathbf{w}_t - \alpha\nabla L_{\text{AVG}}(\mathbf{w}_t))\|^2$.

Applying Lemma 1 to $\mathbf{u}_t$, we have

$$\mathbb{E}[\Delta_{u,t+1}] \leq (1 - \beta_0)\Delta_{u,t} + 2\beta_0^2\sigma^2 + \frac{(1 + \alpha C_{L_{\text{AVG}}})^2}{\beta_0}\|\mathbf{w}_{t+1} - \mathbf{w}_t\|^2. \quad (26)$$

Hence we have

$$\mathbb{E}\left[\sum_{t=0}^{T}\Delta_{u,t}\right] \leq \mathbb{E}\left[\sum_{t=0}^{T}\frac{\Delta_{u,t} - \Delta_{u,t+1}}{\beta_0} + 2\beta_0\sigma^2(T+1) + \sum_{t=0}^{T}\frac{(1 + \alpha C_{L_{\text{AVG}}})^2\eta^2 c_u^2\|\mathbf{z}_{t+1}\|^2}{\beta_0^2}\right]. \quad (27)$$

Defining

$$\mathbf{e}_t = (1 - \beta_1)(\nabla F(\mathbf{w}_t) - \nabla F(\mathbf{w}_{t-1}))$$
$$= (1 - \beta_1)[(I - \alpha\nabla^2 L_{\text{AVG}}(\mathbf{w}_t))\nabla L_{\text{CW}}(\mathbf{w}_t - \nabla_{L_{\text{AVG}}}(\mathbf{w}_t)) \quad (28)$$
$$- (I - \alpha\nabla^2 L_{\text{AVG}}(\mathbf{w}_{t-1}))\nabla L_{\text{CW}}(\mathbf{w}_{t-1} - \nabla_{L_{\text{AVG}}}(\mathbf{w}_{t-1}))],$$

we get

$$\|\mathbf{e}_t\|^2 \leq (1 - \beta_1)^2 L_F^2\|\mathbf{w}_t - \mathbf{w}_{t-1}\|^2. \quad (29)$$

It holds that

$$\mathbb{E}\|\mathbf{z}_{t+1} - \nabla F(\mathbf{w}_t) + \mathbf{e}_t\|^2 \le \mathbb{E}\|(1-\beta_1)(\mathbf{z}_t - \nabla F(\mathbf{w}_{t-1})) + \beta_1((I - \alpha\nabla^2 L_{\text{AVG}}(\mathbf{w}_t; \mathcal{S}_1))\nabla f(\mathbf{u}_{t+1}; \mathcal{S}_2) - F(\mathbf{w}_t))\|^2$$

$$= \mathbb{E}\|(1-\beta_1)[\mathbf{z}_t - \nabla F(\mathbf{w}_{t-1})]$$
$$+ \beta_1[(I - \alpha\nabla^2 L_{\text{AVG}}(\mathbf{w}_t; \mathcal{S}_1))\nabla L_{\text{CW}}(\mathbf{u}_{t+1}; \mathcal{S}_2) - (I - \alpha\nabla^2 L_{\text{AVG}}(\mathbf{w}_t))\nabla L_{\text{CW}}(\mathbf{u}_{t+1})$$
$$\qquad + (I - \alpha\nabla^2 L_{\text{AVG}}(\mathbf{w}_t))\nabla L_{\text{CW}}(\mathbf{u}_{t+1}) - (I - \alpha\nabla^2 L_{\text{AVG}}(\mathbf{w}_t))\nabla L_{\text{CW}}(\mathbf{w}_t - \nabla L_{\text{AVG}}(\mathbf{w}_t))]\|^2$$

$$= \mathbb{E}[(1-\beta_1)^2\|\mathbf{z}_t - \nabla F(\mathbf{w}_{t-1})\|^2]$$
$$+ \beta_1^2\mathbb{E}\|(I - \alpha\nabla^2 L_{\text{AVG}}(\mathbf{w}_t; \mathcal{S}_1))\nabla L_{\text{CW}}(\mathbf{u}_{t+1}; \mathcal{S}_2) - (I - \alpha\nabla^2 L_{\text{AVG}}(\mathbf{w}_t))\nabla L_{\text{CW}}(\mathbf{u}_{t+1})$$
$$\qquad + (I - \alpha\nabla^2 L_{\text{AVG}}(\mathbf{w}_t))\nabla L_{\text{CW}}(\mathbf{u}_{t+1}) - (I - \alpha\nabla^2 L_{\text{AVG}}(\mathbf{w}_t))\nabla L_{\text{CW}}(\mathbf{w}_t - \nabla L_{\text{AVG}}(\mathbf{w}_t))\|^2$$
$$+ 2(1-\beta_1)\beta_1\mathbb{E}[(\mathbf{z}_t - \nabla F(\mathbf{w}_{t-1}))^\top((I - \alpha\nabla^2 L_{\text{AVG}}(\mathbf{w}_t; \mathcal{S}_1))\nabla L_{\text{CW}}(\mathbf{u}_{t+1}; \mathcal{S}_2) - (I - \alpha\nabla^2 L_{\text{AVG}}(\mathbf{w}_t))\nabla L_{\text{CW}}(\mathbf{u}_{t+1}))]$$
$$+ 2(1-\beta_1)\beta_1\mathbb{E}[(\mathbf{z}_t - \nabla F(\mathbf{w}_{t-1}))^\top((I - \alpha\nabla^2 L_{\text{AVG}}(\mathbf{w}_t))\nabla L_{\text{CW}}(\mathbf{u}_{t+1}) - (I - \alpha\nabla^2 L_{\text{AVG}}(\mathbf{w}_t))\nabla L_{\text{CW}}(\mathbf{w}_t - \alpha\nabla L_{\text{AVG}}(\mathbf{w}_t)))]$$

$$\le \mathbb{E}[(1-\beta_1)^2\|\mathbf{z}_t - \nabla F(\mathbf{w}_{t-1})\|^2]$$
$$+ 2\beta_1^2\mathbb{E}\|(I - \alpha\nabla^2 L_{\text{AVG}}(\mathbf{w}_t; \mathcal{S}_1))\nabla L_{\text{CW}}(\mathbf{u}_{t+1}; \mathcal{S}_2) - (I - \alpha\nabla^2 L_{\text{AVG}}(\mathbf{w}_t))\nabla L_{\text{CW}}(\mathbf{u}_{t+1})\|^2$$
$$+ 2\beta_1^2\mathbb{E}\|(I - \alpha\nabla^2 L_{\text{AVG}}(\mathbf{w}_t))\nabla L_{\text{CW}}(\mathbf{u}_{t+1}) - (I - \alpha\nabla^2 L_{\text{AVG}}(\mathbf{w}_t))\nabla L_{\text{CW}}(\mathbf{w}_t - \alpha\nabla L_{\text{AVG}}(\mathbf{w}_t))\|^2$$
$$+ (1-\beta_1)^2\frac{\beta_1}{2}\mathbb{E}\|\mathbf{z}_t - \nabla F(\mathbf{w}_{t-1})\|^2$$
$$+ 2\beta_1\mathbb{E}\|(I - \alpha\nabla^2 L_{\text{AVG}}(\mathbf{w}_t))\nabla L_{\text{CW}}(\mathbf{u}_{t+1}) - (I - \alpha\nabla^2 L_{\text{AVG}}(\mathbf{w}_t))\nabla L_{\text{CW}}(\mathbf{w}_t - \alpha\nabla L_{\text{AVG}(\mathbf{w}_t)})\|^2$$

$$\le (1-\beta_1)^2(1 + \frac{\beta_1}{2})\mathbb{E}\|\mathbf{z}_t - \nabla F(\mathbf{w}_{t-1})\|^2 + 4((1 + \alpha L_{L_{\text{AVG}}})^2 + \sigma^2 + C_{L_{\text{CW}}}^2)\beta_1^2\sigma^2$$
$$+ 4\beta_1(1 + \alpha L_{L_{\text{AVG}}})^2 L_{L_{\text{CW}}}^2\|\mathbf{u}_{t+1} - (\mathbf{w}_t - \alpha L_{\text{CW}}(\mathbf{w}_t))\|^2$$

$$\le (1-\beta_1)(1 - \frac{\beta_1}{2})\mathbb{E}\|\mathbf{z}_t - \nabla F(\mathbf{w}_{t-1})\|^2 + \beta_1^2 C_4\sigma^2 + \beta_1 C_5\Delta_{u,t}.$$

where $C_4 := 4((1 + \alpha L_{L_{\text{AVG}}})^2 + \sigma^2 + C_{L_{\text{CW}}}^2)$, $C_5 := 4(1 + \alpha L_{L_{\text{AVG}}})^2 L_{L_{\text{CW}}}^2$.

It follows that

$$\|\mathbf{z}_{t+1} - \nabla F(\mathbf{w}_t)\|^2 \le (1 + \frac{\beta_1}{2})\mathbb{E}\|\mathbf{z}_{t+1} - \nabla F(\mathbf{w}_t) + \mathbf{e}_t\|^2 + (1 + \frac{2}{\beta_1})\|\mathbf{e}_t\|^2$$

$$\le (1 + \frac{\beta_1}{2})(1 - \beta_1)(1 - \frac{\beta_1}{2})\mathbb{E}\|\mathbf{z}_t - \nabla F(\mathbf{w}_{t-1})\|^2 + (1 + \frac{\beta_1}{2})C_4\beta_1^2\sigma^2$$

$$+ (1 + \frac{\beta_1}{2})C_5\beta_1\Delta_{u,t} + \frac{4}{\beta_1}(1 - \beta_1)^2 L_F^2\|\mathbf{w}_t - \mathbf{w}_{t-1}\|^2 \qquad (30)$$

$$\le (1 - \beta_1)\mathbb{E}\|\mathbf{z}_t - \nabla F(\mathbf{w}_{t-1})\|^2 + 2C_4\beta_1^2\sigma^2$$

$$+ 2C_5\beta_1\Delta_{u,t} + \frac{4}{\beta_1}L_F^2\|\mathbf{w}_t - \mathbf{w}_{t-1}\|^2.$$

Thus,

$$\mathbb{E}\left[\sum_{t=0}^T \Delta_{z,t}\right] \le \mathbb{E}\left[\sum_{t=0}^T \frac{\Delta_{z,t} - \Delta_{z,t+1}}{\beta_1} + 2C_4\beta_1\sigma^2(T+1) + \frac{4L_F^2}{\beta_1^2}\eta^2 c_u^2\|\mathbf{z}_{t+1}\|^2 + 2C_5\sum_{t=0}^T \Delta_{u,t}\right]$$

$$\le \mathbb{E}\left[\sum_{t=0}^T \frac{\Delta_{z,t} - \Delta_{z,t+1}}{\beta_1} + 2C_4\beta_1\sigma^2(T+1) + \frac{4L_F^2}{\beta_1^2}\eta^2 c_u^2\|\mathbf{z}_{t+1}\|^2\right.$$

$$\left. + 2C_5\mathbb{E}\left[\sum_{t=0}^T \frac{\Delta_{u,t} - \Delta_{u,t+1}}{\beta_0} + 2\beta_0\sigma^2(T+1) + \sum_{t=0}^T \frac{(1 + \alpha C_{L_{\text{AVG}}})^2\eta^2 c_u^2\|\mathbf{z}_{t+1}\|^2}{\beta_0^2}\right]\right].$$

$$(31)$$

Combining this with (25), we obtain

$$\frac{\eta c_l}{2} \mathbb{E}\left[\sum_{t=0}^{T} \|\nabla F(\mathbf{w}_t)\|^2\right] \leq F(\mathbf{w}_0) - F_* - \sum_{t=0}^{T} \frac{\eta c_l}{4} \|\mathbf{z}_{t+1}\|^2 + \sum_{t=0}^{T} \frac{\eta c_u}{2} \Delta_{z,t}$$

$$\leq \Delta_F + \frac{\eta c_u}{2} \sum_{t=0}^{T} \left[\frac{\Delta_{z,t} - \Delta_{z,t+1}}{\beta_1} + 2C_5 \frac{\Delta_{u,t} - \Delta_{u,t+1}}{\beta_0}\right]$$

$$+ \frac{\eta c_u}{2} [2C_4 \beta_1 + 4C_5 \beta_0] \sigma^2 (T+1) \qquad (32)$$

$$+ \left[\frac{\eta c_u}{2} \left(\frac{4L_F^2 \eta^2 c_u^2}{\beta_1^2} + \frac{2C_5(1+\alpha C_{L_{\mathrm{AVG}}})^2 \eta^2 c_u^2}{\beta_0^2}\right) - \frac{\eta c_l}{4}\right] \sum_{t=0}^{T} \|\mathbf{z}_{t+1}\|^2.$$

Due to the setting

$$\eta \leq \left\{\frac{\sqrt{c_l}\beta_1}{4L_F \sqrt{c_u^3}}, \frac{\sqrt{c_l}\beta_0}{4(1+\alpha C_{L_{\mathrm{AVG}}})\sqrt{C_5 c_u^3}}\right\}, \qquad (33)$$

we have

$$\left[\frac{\eta c_u}{2} \left(\frac{4L_F^2 \eta^2 c_u^2}{\beta_1^2} + \frac{2C_5(1+\alpha C_{L_{\mathrm{AVG}}})^2 \eta^2 c_u^2}{\beta_0^2}\right) - \frac{\eta c_l}{4}\right] \leq 0. \qquad (34)$$

Thus,

$$\mathbb{E}\left[\frac{1}{T+1} \sum_{t=0}^{T} \|\nabla F(\mathbf{w}_t)\|^2\right] \leq \frac{2\Delta_F}{\eta c_l T} + \frac{c_u}{c_l}\left[\frac{\mathbb{E}[\Delta_{z,0}]}{\beta_1 T} + 2C_5 \frac{\mathbb{E}[\Delta_{u,0}]}{\beta_0 T}\right] + \frac{c_u}{c_l}[2C_4\beta_1 + 4C_5\beta_0]\sigma^2.$$

$$(35)$$

With $\beta_0 = O(\frac{1}{\sqrt{T}})$, $\beta_1 = O(\frac{1}{\sqrt{T}})$ and $\eta = O(\frac{1}{\sqrt{T}})$,

$$\mathbb{E}\left[\frac{1}{T+1} \sum_{t=0}^{T} \|\nabla F(\mathbf{w}_t)\|^2\right] \leq O(\frac{1}{\sqrt{T}}). \qquad (36)$$

$\square$

