# OpenReview forum: "Compositional Training for End-to-End Deep AUC Maximization"
_ICLR.cc/2022/Conference — ICLR 2022 Spotlight_

### Official Review · Reviewer_qWty · 2021-10-31

**Correctness:** 3
**Technical Novelty And Significance:** 3
**Empirical Novelty And Significance:** 3
**Recommendation:** 8
**Confidence:** 3

**Main Review:**

The paper includes a good summary of the related work and the contributions. The proposed compositional approach to the deep AUC maximisation problem seems interesting and novel. It uses a reformulation of the optimised loss function that contains three different terms which jointly optimize the feature representation and the classification performance. A primal-dual stochastic algorithm is proposed that optimises the proposed cost function and an informal argument for its convergence is given. The reported experimental results are quite convincing, both for the computer vision tasks and for the medical imaging tasks, however a proper comparison of the experimental results with other state-of-art methods is missing.


**Summary Of The Paper:**

The paper addresses the problem of how to perform end-to-end training for deep AUC ("area under ROC curve") maximisation. This is a challenging problem as this yields a harder optimisation problem than the standard CE minimisation. Existing approaches use two stage optimisation approaches where the first stage optimises a traditional loss function (e.g. the CE loss) and the second stage fine-tunes the network by optimising the AUC loss. The proposed approach is termed compositional deep AUC maximisation which can be trained end-to-end. The paper furthermore proposes an efficient stochastic optimisation algorithm solving compositional deep AUC maximisation problem. It has been evaluated on standard benchmark computer vision and medical imaging classification datasets.

**Summary Of The Review:**

This is an interesting paper that proposed a number of technical advances (formulation of the compositional loss function and its optimization) with reasonable results, albeit there are weaknesses in the experimental evaluation.

---

> ### Author Response · Authors · 2021-11-22
> **Regarding Reviewer qWty‘S Concerns**
>
> Thanks for your useful comments! Below we address your question.
>
> **Q1**: A proper comparison of the experimental results with other state-of-art methods.
> - **Response**:  Thank you! The focus of this paper is to compare with the state of the art AUC maximization method, e.g., Yuan et al. 2020. Please note that Yuan et al. 2020 have already compared with some state-of-the-art methods, e.g., the focal loss, and showed that their AUC maximization method is better. In this paper, we showed that our method is better than Yuan et al.’s method.

---

### Official Review · Reviewer_LxEz · 2021-10-31

**Correctness:** 3
**Technical Novelty And Significance:** 3
**Empirical Novelty And Significance:** 3
**Recommendation:** 8
**Confidence:** 3

**Main Review:**

Strengths:
1. The paper provides good explanation on their newly proposed compositional loss and training method.
2. The paper performs extensive experiments to demonstrate the effectiveness of their method.
3. This paper is well written and organized.

Weaknesses:
Overall, this paper is a good work. But I do have several questions and concerns:
1. The author hypotheses that using CE loss, different examples roughly have equal weights regardless which classes they belong to. And it is one of the key reasons that using CE loss has advantage over raw AUC loss. It would be great to show that the compositional loss can help achieve the equal weights property empirically to validate this hypothesis in the paper.
2.  In the experiments section, the authors list AUC for multi-class classification tasks (e.g. CIFAR10 and CIFAR100), I wonder how the AUC is computed for these multi-class tasks. The details are missing in the paper.
3. For medical image datasets, since the authors only report the results from a single run, it may not be very convincing to show the improvement. I would suggest the authors either report results from multiple runs or report the evaluation noise level (i.e train same model several times and perform evaluation on the test data, AA test to understand the eval noise), so that we can make sure the improvement comes from the proposed method.
4. In figure 4, there are several bump ups in the convergence curves. What are the reasons of these bumps? Are they because a learning rate scheduling or something else? The detailed explanations are missing.
5. In table 2, the authors mention that they tune the inner gradient steps k \in {1,2,3} for the left table while keep k = 1 for the right table. However, the detailed tuning strategy is missing.  Does the algorithm tries out 1,2,3 and pick up the optimal one automatically during each iteration?
6. In the runtime analysis, it would also be useful to report what hardware is used for testing, since different hardware may results different running time.


**Summary Of The Paper:**

This work proposes a new compositional loss function that aims to help deep AUC maximization. Specifically, the outer function of the newly proposed loss function is a surrogate loss for AUC, i.e. AUC square loss; while the inner component can be understood as to facilitate the convergence of CE loss. The authors explain their compositional loss through Taylor expansion theoretically and the lens of feature purification and classifier robustification intuitively. Furthermore, they demonstrate the difference between their proposal and the linear combination of AUC and CE loss from both mathematical perspective and empirical perspective. Experimental wise, this paper has done extensive studies on different dataset ranging from popular natural image benchmarks to several medical image datasets. Though the improvement is rather limited on some cases, CT(AUC) outperforms all the other methods in all cases.

**Summary Of The Review:**

Overall, this paper is well written and explained. I have listed several questions and concerns in the Main Review.

---

> ### Author Response · Authors · 2021-11-22
> **Regarding Reviewer LxEz’s Concerns**
>
> Thanks for your useful comments! Below we address your questions.
>
> **Q1**: It would be great to show that the compositional loss can help achieve the equal weights property empirically to validate this hypothesis in the paper.
> - **Response**:  The equal weight property is ensured by the inner gradient step using the CE loss in our compositional loss. We argue that our method enjoys the advantage of the CE loss over the raw AUC loss due to the equal weights for all examples for better feature learning. This has been validated in Figure 3. In addition, in Appendix A.6 we have verified that the compositional training can push both the CE loss and the AUC loss to be smaller than a baseline that optimizes the linear combination of the CE loss and the AUC loss.
>
> **Q2**: How the AUC is computed for these multi-class tasks.
> - **Response**:  We are sorry for the confusion! As we mentioned in the paper “For AUC maximization, we construct imbalanced binary versions of these datasets by varying the imbalanced ratios (the ratio of positive examples to the total number of training examples) similar to Yuan et al. 2020”. Hence for all benchmark datasets including CIFAR10 and CIFAR100, we construct their versions for binary classification. For example, we re-label the first 50 classes as positive class and the last 50 classes as negative class. Then, we construct the imbalanced set by randomly removing some positive samples from the training set. For the validation/testing set, we label them using the same two classes. The final AUC is calculated and reported for the binary classification tasks.
>
> **Q3**: Multiple runs on medical image datasets.
> - **Response**:  Thanks for the suggestions! We re-run experiments on all medical datasets for three runs, e.g., using different model initializations. The new results are reported in the following table. Please refer to the response of R1 for details.
> |  Methods |   Melanoma  |   CheXpert  |    DDSM+    |   PatchCam  |
> |:--------:|:-----------:|:-----------:|:-----------:|:-----------:|
> |  Methods |     AUC     |     AUC     |     AUC     |     AUC     |
> |    CE    | 0.879±0.008 | 0.892±0.001 | 0.949±0.001 | 0.869±0.007 |
> |    AUC   | 0.868±0.006 | 0.899±0.002 | 0.929±0.001 | 0.868±0.006 |
> |  AUC-CE  | 0.880±0.005 | 0.902±0.002 | 0.957±0.001 | 0.868±0.005 |
> |  TS-DRW  | 0.878±0.007 | 0.900±0.002 | 0.942±0.003 | 0.867±0.006 |
> |  TS-DEC  | 0.877±0.005 | 0.897±0.001 | 0.941±0.001 | 0.869±0.009 |
> | **CT (AUC)** | **0.900±0.002** | **0.909±0.003** | **0.981±0.001** | **0.891±0.003** |
>
> **Q4**: Several bump ups in the convergence curves in figure 4.
> - **Response**: In the experiments, we use stagewise learning rate scheduling. For example, we use initial step size 0.1 and decrease it by 10 times at 50% and 75% of total training time for all benchmark experiments. Thus, these bump ups indeed occur as the learning rate is decayed.
>
> **Q5**: The detailed tuning strategy for inner gradient steps k.
> - **Response**: Sorry for the confusion! We use the same value of k across all iterations but tune k in the range {1,2,3} on the validation set and report the best results with the best  k on testing sets.
>
> **Q6**: Report what hardware is used for experiments.
> - **Response**: We have included some details of training configuration in Appendix A.2 as mentioned in the paper, which has reported the hardware used for different data.  In particular, all benchmark datasets are experimented by NVIDIA GTX-2080Ti and four medical datasets,i.e., CheXpert, Melanoma, DDSM+ and PatchCam, are experimented by NVIDIA V100.

---

### Official Review · Reviewer_TF22 · 2021-11-02

**Correctness:** 3
**Technical Novelty And Significance:** 3
**Empirical Novelty And Significance:** 3
**Recommendation:** 6
**Confidence:** 3

**Main Review:**

While I cannot speak to the proposed optimization approach, I feel the following are the major strengths and weaknesses of the paper:

Strengths:

- The authors present the first end-to-end training framework for AUC maximization (previous approaches required pretraining e.g. using standard CE loss first for good results)

- The authors present an optimization method for the new loss that has similar rate of convergence as standard SGD

- Experiments are presented using a large number of natural and medical image public datasets.

- Generally the results show consistent improvement over AUC / CE - only loss, a standard linear combination of AUC + CE losses, and 2 other methods for maximizing AUC loss.

Weaknesses:

- My primary concern is some missing details in the experiments, which makes it difficult to accept the results at face value. While some of the datasets do mention a separation of training/validation/test, several datasets only discuss having either only validation or test set (eg. 3 out of the 4 medical datasets). Thus, it is unclear how the parameters for the different methods are tuned - even though all the methods are tuned on the same dataset partitions, if tuned on the set used for reporting results, this will not capture the generalization capability of the methods and thus it is difficult to assess which is really best. Please clarify the parameter tuning process.

- I am confused about the DDSM+ dataset, where the authors state they have combined the DDSM with CBIS-DDSM datasets. This doesn't make sense to me, as my understanding is that CBIS-DDSM is an updated/standardized version of the DDSM, i.e. CBIS-DDSM is derived from the DDSM and contains the same patient data (also, the authors do not have the correct citation for CBIS-DDSM, which should be: Lee et al., A curated mammography data set for use in computer-aided detection and diagnosis research, 2017). Thus depending on how the data is partitioned, the same patient data will be in both training/testing sets, which may partially explain the extremely high AUC results. Furthermore, I am not sure how there are so many images (55K training and ~14K testing), as each dataset has approximately 2600 patients (again, the same patients), where each should have ~4 associated mammogram images (left/right and MLO/CC views). Please clarify the use of the mammography datasets.

- The authors test on all the same datasets as those presented in the paper by Yuan et al., 2020, which introduced the AUCM loss. However, there are no comparisons made to the results from this paper, which seem to report higher AUC results for many of the datasets (eg. CheXpert, Melonoma, PatchCam). While the authors compared to AUCM loss, it was for training from scratch, not using the same training framework as proposed in Yuan et al. It would make sense that the authors should be comparing to the results in this paper if directly comparable, or running their own comparison to this method.

- Regarding the ablation studies on algorithmic design choices for beta and k in Sec. 4.1, I feel like we do not learn much new from this, as it simply verifies that it is better to tune hyperparameters than to not tune them.

Minor comments:

- Although understandable, would be good to define what "imratio" means
- The convergence curve results don't really seem to fall under "ablation" study, as nothing is being ablated...consider moving to other section or renaming 4.1 for clarity of presentation
- p.2: "presentations" --> representations


**Summary Of The Paper:**

This paper presents a new loss function for end-to-end training of deep networks for AUC maximization.  The proposed compositional loss combines the standard cross-entropy (CE) loss within the AUC loss such that the AUC is maximized for good classification while the CE portion of the loss helps to learn robust features. An optimization method is presented and is shown to have the same convergence rate as standard SGD. The proposed method is tested on 4 benchmark and 4 medical image datasets and compared against related losses, showing consistent improvement in AUC.


**Summary Of The Review:**

My recommendation is based on the introduction of the novel end-to-end learning approach for AUC maximization and the presented empirical results on 8 different datasets that show general improvement compared to other AUC maximization / CE approaches. However, my enthusiasm was dampened by the unclear experimental setup for parameter tuning/use of validation sets and lack of comparison to highly related work.

---

> ### Author Response · Authors · 2021-11-22
> **Regarding Reviewer TF22’s Concerns**
>
> Thanks for your useful comments! Below we address your questions.
>
> **Q1**: Please clarify the parameter tuning process.
> - **Response**: We apologize for missing details! The Appendix A.2 has included some training configurations. For all datasets, we use the train/val split to do cross-validation for parameter tuning, except CheXpert as explained below. For the benchmark datasets, we use 19k/1k, 45k/5k, 45k/5k. 4k/1k training/validation split on CatvsDog, CIFAR10, CIFAR100, STL10, respectively. For melanoma dataset, we use 70/10/20 split for train/val/test as reported in the paper. For PatchCam, we use their official validation set for tuning parameters, which includes about 37k images with balanced positive and negative samples. For DDSM+, we tune the parameters on 10% data sampled from the training set.  For CheXpert, since the official testing set is not released and it will take a long time to evaluate all methods on the official testing data,  hence, we evaluate different methods only based on the official validation set with parameters tuned according to this set.
> - To make the experiment on Chexpert consistent with other datasets, we re-run all methods on CheXpert by following the same cross-validation procedure, i.e.,  by sampling 10% training data based on patient ID as the validation set to tune parameters and then we report the average scores of five diseases on the testing set (i.e., the official validation set as a testing set).
> - For all medical datasets, we re-run all experiments and report the average performance over three runs. We use batch size=32 except for PatchCam that is 64, initial learning rate=0.1 and weight decay=1e-5. We train Melanoma for 12 epochs, CheXpert for 2 epochs, DDSM+ for 5 epochs and PatchCam for 5 epochs. The learning rate is decayed at 50%, 75% of total training iterations by 10 times. The results are summarized in the following table.
>
> |  Methods |   Melanoma  |   CheXpert  |    DDSM+    |   PatchCam  |
> |:--------:|:-----------:|:-----------:|:-----------:|:-----------:|
> |  Methods |     AUC     |     AUC     |     AUC     |     AUC     |
> |    CE    | 0.879±0.008 | 0.892±0.001 | 0.949±0.001 | 0.869±0.007 |
> |    AUC   | 0.868±0.006 | 0.899±0.002 | 0.929±0.001 | 0.868±0.006 |
> |  AUC-CE  | 0.880±0.005 | 0.902±0.002 | 0.957±0.001 | 0.868±0.005 |
> |  TS-DRW  | 0.878±0.007 | 0.900±0.002 | 0.942±0.003 | 0.867±0.006 |
> |  TS-DEC  | 0.877±0.005 | 0.897±0.001 | 0.941±0.001 | 0.869±0.009 |
> | **CT (AUC)** | **0.900±0.002** | **0.909±0.003** | **0.981±0.001** | **0.891±0.003** |
>
> **Q2**: Please clarify the use of the mammography datasets.
> - **Response**: Thanks for raising this concern!  A short answer to your concern is that there is no overlap in terms of patients in training and testing sets. The DDSM+ is slightly different from the standard version of CBIS-DDSM or DDSM. We actually use the dataset from [1] constructed by Eric A. Scuccimarra, which consists of 55k images for training and 15k images for testing.  The details about the dataset construction can be found here [2, 4, 5]. For constructing DDSM+, the positive samples (cancer cases) are from CBIS-DDSM and negative samples (normal cases) are from DDSM. To increase the size of the training data, the author applies offline data augmentation and adds multiple augmented copies to the dataset. In particular, each image (ROI) is randomly cropped three times into 598x598 images, with random flips and rotations, and then the images are resized down to 299x299. For the testing set, the same augmentation is also applied and thus the imbalance ratios remain the same as the training set.  The imbalance ratio is about 13% in both training and testing sets (e.g., there is a typo regarding this ratio for testing set in the paper). The train/test split in terms of patient ID follows the CBIS-DDSM split, which do not have any overlap. We will add these references in the revision and also correct the citation for CBIS-DDSM.  Please also note that this dataset (DDSM+) has also been used for some recent works [4, 5].
>
> **Reference**:
> - [1] ddsm-mammography. https://www.kaggle.com/skooch/ddsm-mammography.
> - [2] ddsm-mammography. https://github.com/escuccim/mias-mammography/blob/master/Report.md
> - [3] U.S. Breast Cancer Statistics. https://www.breastcancer.org/symptoms/understand_bc/statistics.
> - [4] Zheng, Yan, et al. "Top-rank convolutional neural network and its application to medical image-based diagnosis." Pattern Recognition 120 (2021): 108138.
> - [5] Fulton, Lawrence, et al. "Deep Vision for Breast Cancer Classification and Segmentation." Cancers 13.21 (2021): 5384.

---

> ### Author Response · Authors · 2021-11-22
> **Regarding Reviewer TF22’s Concerns**
>
>
> **Q3**: Regarding the comparisons with Yuan et al.
> - **Response** : We apologize for any confusion on this part!  First, we would like to point out that for the baselines TS-DEC, TS-DRW, we indeed use the AUCM loss and the PESG optimizer (during the second stage) as proposed in Yuan et al 2020. In particular, for these two baselines we use CE loss to pretrain the feature encoder in the first stage and then we fine-tune the last layer (for TS-DEC) or all layers (for TS-DRW) in the second stage.  This two-stage training method is indeed used by Yuan et al. 2020, which reported that a two-stage method (with the reference to Kang et al 2019 that proposes TS-DEC) is used for medical datasets without too much detail. Hence, we assure you that the baseline TS-DEC or TS-DRW implements the method proposed by Yuan et al. to our best.
> - Second, as we mentioned in the paper (the paragraph above section 4.1) there are some differences in experiment setup between Yuan’s work and this work: 1) we use sigmoid activation in the last output layer for all experiments instead of the l2 normalization used in their paper.  2) For benchmark datasets, our results are the averages of different runs on the same imbalanced training datasets using different model initializations while the randomness of Yuan’s results comes from different random imbalanced training datasets.  3) Due to the limited resources, we are not able to run same experiments using large image resolutions on medical datasets, e.g, 384x384, 512x512, and thus we resize all images to smaller size, e.g., 224x224.
>
> **Q4**: Regarding the ablation studies on algorithmic design choices for beta and k in Sec. 4.1.
>
> - **Response**:  We consider these ablation studies to be necessary as they verify our algorithmic choices. They not just show that tuning hyper-parmeters is better than not tuning them but also verify some of our algorithmic choices. In particular, we show that using $\beta_0<1$ is better than a naive version with $\beta_0=1$, and also show that using two different sets $S_1\neq S_2$ for computing the stochastic gradient estimator is better than using the same set for generalization. In fact, these experiments are following suggestions of reviewers of the previous version of this paper.
>
> **Q5**: Other minor comments, such as typos.
>
> - **Response**: Thanks for the suggestions! We have added the explanation of **imratio** in the appendix and fixed the typos.

---

### Author Response · Authors · 2021-11-22
**Response to all reviewers**

Dear all reviewers,

Thank you for reading our responses! We have also updated our manuscript to address the concerns raised by all reviewers. The changes made in main section and appendix in the revised draft are marked by red color. Please let us know if you have any further questions. Thank you!

Authors

---

### Decision · Program_Chairs · 2022-01-20

**Decision:**

Accept (Spotlight)

**Comment:**

I recommend this paper to be accepted. All reviewers are in agreement that this paper is above the bar.